# Exploring Spatial-Temporal Coupling and Its Driving Factors of Green and Low-Carbon Urban Land Use Efficiency and High-Quality Economic Development in China

Lina Peng [1,2,3,4], Juan Liang [1,2], Kexin Wang [1,2], Wenqian Xiao [1,3], Jian Zou [1,3], Yuxuan Hong [1,4] and Rui Ding [1,2,3,4],*

1. College of Big Data Application and Economics (Guiyang College of Big Data Finance), Guizhou University of Finance and Economics, Guiyang 550025, China; 15215244950@mail.gufe.edu.cn (L.P.); liangjuan@mail.gufe.edu.cn (J.L.); 844135652@mail.gufe.edu.cn (K.W.); xiaowenqian2000@mail.gufe.edu.cn (W.X.); 20221172111003@mail.gufe.edu.cn (J.Z.); 20221272111075@mail.gufe.edu.cn (Y.H.)
2. Guizhou Collaborative Innovation Center of Green Finance and Ecological Environment Protection, Guizhou University of Finance and Economics, Guiyang 550025, China
3. Artificial Intelligence and Digital Finance Lab, Guizhou University of Finance and Economics, Guiyang 550025, China
4. Guizhou University of Finance and Economics Regional Economic High-Quality Development Research Provincial Innovation Team, Guiyang 550025, China
* Correspondence: 201801162@mail.gufe.edu.cn; Tel.: +86-186-2883-7118

**Abstract:** Green and low-carbon use of urban land (GLUUL) and high-quality economic development (HED) are two closely linked and mutually reinforcing systems, and their coordinated development is of great theoretical and practical significance to the realization of green and sustainable urban development. Based on theoretical analysis, this paper used data from 2005 to 2020 to measure GLUUL efficiency and HED level and their coupling coordination degree (CCD) successively of 282 cities in China, and then analyzed in-depth the main factors affecting CCD and its spatial–temporal heterogeneity using the GTWR model. This study found that (1) GLUUL efficiency and HED levels are increasing with different trends, and the development is unbalanced. High-value cities in the two systems show a staggered distribution pattern. (2) The CCD of the two was dominated by primary and intermediate coordination types, and the overall became increasingly coordinated, with the "intermediate coordination—advanced development" type having the highest representation. (3) There is a gradual convergence of CCD spatial differences, showing an overall spatial distribution pattern that is "high in the northwest and southeast, low in the central area". (4) The influence degree and direction of different factors on CCD are distinguishing. The positive impact of industrial structure upgrading (Isu) is obviously greater than other factors, which has the strongest effect on the industrial corridor along the Yangtze River and the Beijing–Tianjin–Hebei region. The findings can offer insightful recommendations for promoting sustainable development in China and similar developing countries and regions.

**Keywords:** green and low-carbon land use; urbanization; high-quality economic development; coupling coordination; influencing factors; spatial–temporal heterogeneity

## 1. Introduction

In 2007, the 13th Conference of the Parties of the United Nations Framework Convention on Climate Change adopted the "Bali Action Plan", which proposed the concepts of a "low-carbon economy" and "clean energy" and called on all countries to take measures to address global climate change. Since then, more and more attention has been placed on how to achieve green and low-carbon transformation of land utilization and raise the efficiency of resource use [1]. Urbanization is an irresistible trend in the 21st century, and most of

the urban expansion will take place in developing countries in the next few decades [2,3]. China, being the most populous developing nation globally, has witnessed an unprecedented urbanization process since the 1980s. In 1987, the rate of China's urbanization was only 17.9%. However, by 2021, it reached 64.72%, with an annual growth rate of over 1% [4]. This exceeds the world average for the same period. The rapid urbanization evolution created augmented pressure on land resources, environmental pollution, climate change, and other problems, posing severe challenges to regional sustainable development. To this end, China has put forward the strategy of "new-type urbanization" and the five development concepts of "innovation, coordination, green, openness and sharing" to further promote GLUUL (green and low-carbon use of urban land) and HED (high-quality economic development). GLUUL can increase the utilization efficiency and ecological benefit of urban land resources and reduce the adverse effects on the environment, while HED needs more high-quality land resources to facilitate the adjustment and upgrading of the industrial constitution to create innovation-driven cities and other aspects. Therefore, in the process of urbanization, it has important practical significance to coordinate GLUUL and HED.

GLUUL means that social, economic, and ecological gains should be as large as possible, while the environmental losses should be as small as possible in urban land systems under certain resource inputs. Specifically, GLUUL emphasizes that the balance and stability of the ecosystem should be fully considered in the process of land use, avoiding the destruction of urban wetlands, forests, and grasslands. Secondly, GLUUL focuses on reducing environmental pollution. By reducing the emission of pollutants such as wastewater, exhaust gas, and solid waste produced during land use, it reduces the pollution of soil, water bodies, and the atmospheric environment. In addition, it also stresses the lowering of greenhouse gas emissions through rational planning of land use layout, construction of low-carbon transport systems, use of clean energy, *etc.*, to reduce $CO_2$ emissions from urban social production and living. Relevant studies predominantly focused on three aspects: first, urban planning and design. Urban land use efficiency can be effectively improved by appropriating urban form and pursuing optimal city size. Some studies have examined urban forms using landscape indicators to quantify sprawl, complexity, and agglomeration, and found that irregular urban forms have a negative impact on urban land use efficiency, while compact and agglomerated urban forms can improve it [5]. The same evidence comes from remote sensing data, which found that an urban form characterized by high patch density and large area, although conducive to improving land use efficiency in large cities, is not conducive to improving land use efficiency in small cities [6]. The second aspect is industrial transformation and agglomeration. It can improve the input–output efficiency of urban land resources by upgrading the industrial structure and optimizing the spatial layout of industries. Studies have shown that there is a synergistic effect of interactive growth between urban land use efficiency and industrial transformation [7]. Industrial specialized agglomeration has an inverted U-shaped relationship with urban green space use efficiency, while industrial diversified agglomeration has a positive effect [8]. Third, the establishment of low-carbon cities and communities [9]. The construction of low-carbon cities and green communities promotes the coordination of economic development and green land use in urban areas. When examining the effectiveness of China's low-carbon city pilot policy, a study found that the policy promoted land green use efficiency in the eastern and western regions, as well as in growing resource-based cities [10]. Similarly, smart city construction can significantly improve urban land green use efficiency through the development of the information industry and regional innovation capacity. This impact is more pronounced in mega and above cities [11]. Urban land use efficiency is evaluated mainly through DEA (Data Envelopment Analysis) and SBM (Slacks-Based Measure) models [12,13]. These models comprehensively consider non-desired outputs, such as urban wastewater and urban exhaust gas emissions, from the perspective of inputs and outputs by establishing environmental constraints [14–16]. Indicator evaluation is another representative method that includes both composite and single indicators. For example,

Ustaoglu. E and Aydınoglu (2020) evaluated the suitability of urban construction sites using various indicators, such as geographic quality, accessibility, built-up area conditions, urban greenery, and amenities [17]. He et al. (2019) used the value added per square kilometer of secondary and tertiary industries as an indicator of urban land use efficiency from a sustainability perspective [6].

As the carrier of urban economic activities, the limited supply of land determines that improving urban land use efficiency is an inherent requirement for sustainable economic development. Urban land use efficiency is closely related to the level of economic development [18,19]. Studies have shown that regional economic integration in metropolitan areas can promote the optimal allocation of resources for improving urban land use efficiency during the socioeconomic transformation process [20]. In turn, urban land use efficiency can impact economic development through the economic scale effect, economic structure optimization effect, and economic quality improvement effect [21]. With China's economic growth slowing down in recent years and resource and environmental constraints tightening, the term "high-quality economic development" was first introduced at the 19th National Congress of the Communist Party of China (CPC) [22]. Compared with the approach of rapid economic growth at the cost of resource consuming and environmental contamination, the HED mode, which upholds the five concepts of innovation, coordination, green, openness, and sharing, pays more attention to sustainability and stability. Researchers have extensively studied the fundamental meaning, index system construction, and development level measurement of HED [23,24]. Then, the regional disparity and spatial–temporal differentiation are studied to point out the striking regional development disparity [25,26], locate the high and low distribution of HED levels, and propose coordinated development countermeasures. Further, the research focus turns to the driver analysis of HED [27,28]. With the continuous practice of HED, research on coordinated relationships between economic or social aspects and HED has been given more attention, including scientific and technological innovation [29], digital economy [30], green finance [31], ecological protection [32], etc.

The literature review indicates that research results on urban land use and HED are very rich, but there is still some room for expansion. In terms of land use, few researchers integrate both "environmentally friendly" and "low carbon" into urban land use systems, while GLUUL is an inevitable requirement for sustainable development. From a research perspective, few scholars pay attention to the coupling coordination of land and economic systems during urbanization, while investigating the relationship between GLUUL and HED has a realistic value in promoting the coordinated development of a "resource-economy-environment". In the aspect of measurement methods, previous studies mostly used econometric models to analyze influencing factors, which failed to deeply explore their spatial–temporal dynamic changes [33,34]. Considering this, this article aims to provide a holistic perspective on GLUUL and HED and their coordination relationship and proposed a systematic and coherent framework for in-depth analysis (Figure 1). Taking 282 cities in China as research objects, this paper uses data from 2005 to 2020 to gradually study the spatial–temporal evolution characteristics of GLUUL efficiency and HED level and their coordination relationship and uses the GTWR model to reveal the dynamic evolution of the main factors affecting CCD, and then, some valuable policy references are proposed.

The innovations and research value of this paper consist of the following. (I) In terms of research perspective, urban land use efficiency is explored from the environmentally friendly and low-carbon perspective. (II) In the aspect of theoretical innovation, GLUUL and HED are incorporated into the coupling theory analysis, and the mutual promotion logic between them is explained in detail. (III) In terms of path innovation, the GTWR model is utilized to objectively examine the factors influencing CCD, and the effects of influencing factors in different times and spaces are deeply analyzed, which can provide a spatially dynamic outlook to facilitate coordinated development between GLUUL and HED.

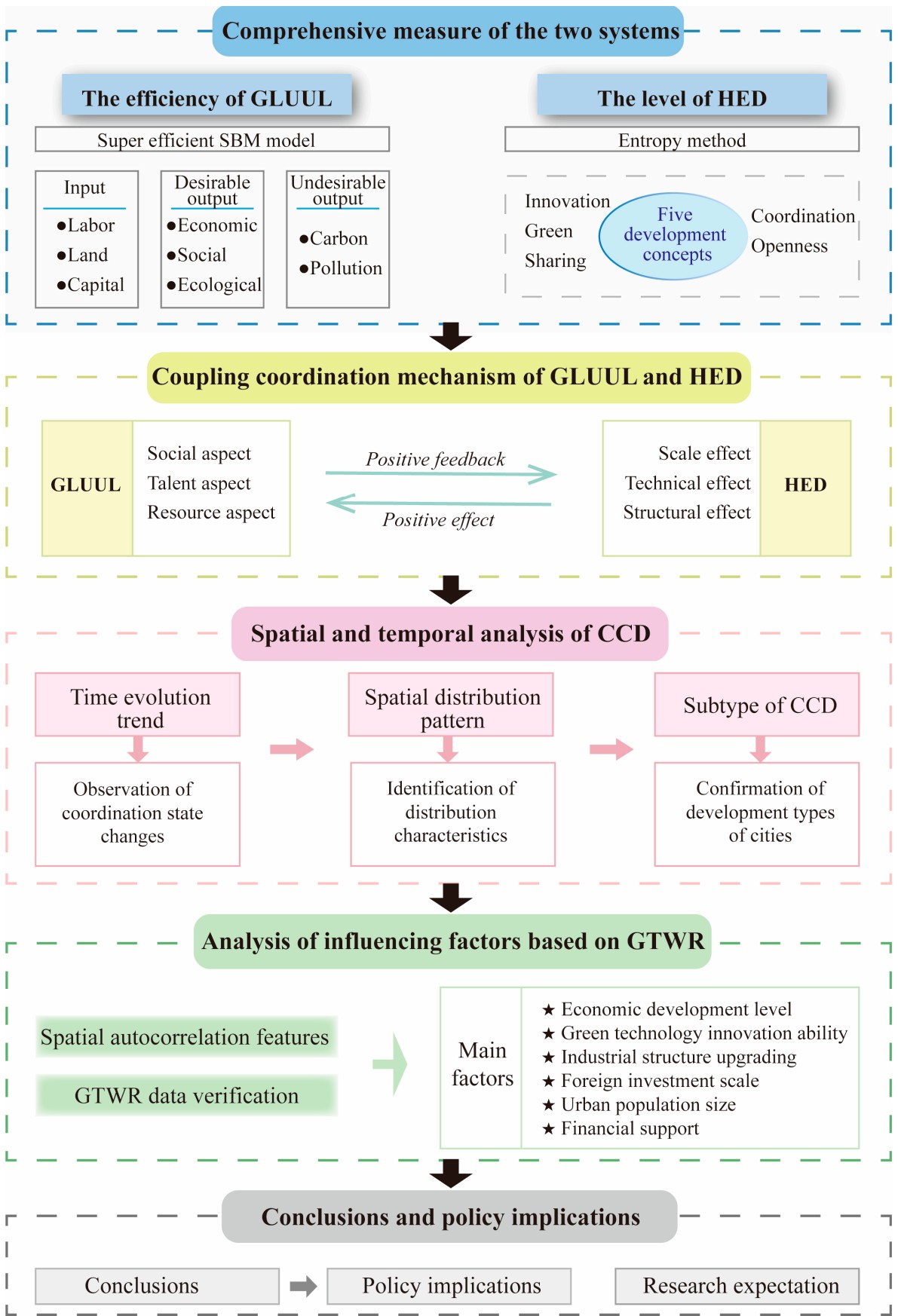

**Figure 1.** The research framework of this study.

## 2. Theoretical Mechanism

Coupling theory is one of the theories widely used to explore the coordination relationship between systems in the process of sustainable development [35]. Coupling is the description of the interaction between two or more systems [36]. GLUUL and HED, as two complex systems that influence each other, can reach a harmonious state through mutual promotion. Based on the interaction between GLUUL and HED, the coupling mechanism analysis framework of GLUUL and HED was constructed in this paper.

On the one hand, HED has effects on GLUUL. The region with a higher HED level tends to have a larger economic scale, higher technical level, and more reasonable industrial structure. Thus, HED produces three effects on GLUUL. (I) Scale effect. Studies have shown that the emergence of a scale economy has linked large-scale production factories and industrial clusters to the reduction in input factor costs [37]. This economic model enhances economic efficiency per unit of land, reduces the cost of resource input, and achieves greater capacity scale, thus promoting GLUUL. (II) Technical effect. Areas with a greater level of economic development possess greater capability and willingness to increase investment in research and development. In the pro-environment field, technological innovation offers the potential to enhance the financial gain from each plot of land, as well as to lessen the amount of carbon emissions [38,39], thus motivating GLUUL. (III) Structural effect. The transformation of the economic structure brings a transition from resource-intensive industries to technology-intensive ones [40], forming several "low pollution and low energy consumption" industries or enterprises [41], which tend to use land with the "low input, high yield and low pollution" mode [7], thus enhancing GLUUL.

On the other hand, as the cornerstone of urban economic activity, the core of GLUUL is that the social, economic, and ecological gains should be as large as possible, while the environmental losses should be as small as possible in the urban land system under the input of certain production factors [42]. HED, moreover, stresses more efficient and ecological economic development. Therefore, GLUUL, promoted by optimizing input structure and reducing environmental losses, can provide carrying space for HED, further influencing HED from three aspects as follows. (I) Social aspect. Firstly, in the economic spatial layout, integrating urban land use can guide moderate population agglomeration, develop the industrial clusters, and allocate urban resources rationally, which is a prerequisite for HED [43]. Secondly, in the urban interior green space, the excessive expansion of urban construction land will lead to a serious occupation of city green areas and the deterioration of climate, temperature, and other living conditions [44], which is not conducive to the urban ecological environment. Finally, the carbon emissions and environmental contamination generated by the irrational use of urban land can significantly impact the physical and mental health of citizens [45], ultimately reducing their quality of life and exacerbating regional pollution. Moreover, it will affect the health of workers psychologically and physically, and it is detrimental to the promotion of a healthy and harmonious society. (II) Talent aspect. The irrational use of land leads to the deterioration of the urban living environment, which will not only increase the cost of public health but also increase the risk of net loss of the social labor force [46]. In addition, as people with higher education have higher requirements for life quality [47], it will also affect the introduction of high-quality talents and retention of original high-tech talents. Thus, the accumulation of human capital, which is one of the basic elements of a region's economic development, will be further affected. (III) Resource aspect. Natural resources and productive resources are the material guarantee for development. GLUUL requires that the social, economic, and ecological output should be maximized under a certain resource input. In other words, at a certain output, GLUUL needs minor resource inputs. Thereby it can achieve the goal of saving resources and provide more reliable material guarantees for social and economic development.

To sum up, HED can positively promote GLUUL through scale, technology, and structure effects, while the improvement of GLUUL positively feeds back to HED through the social effect, talent effect, and resource effect. In this cycle, the two systems can achieve

a coupling and coordination state through mutual promotion, and finally, high levels of GLUUL and HED can be reached.

## 3. Research Design

### 3.1. Index System Construction

This paper focuses on urban land, which comprises residential land, land for public administration and public service facilities, land for commercial service facilities, industrial land, land for logistics and warehousing, land for roads and transport facilities, land for public utilities, and land for green areas and squares within the city. It is important to note that rural and agricultural land within the city is not included in the scope of this study. In accordance with the connotation of GLUUL proposed above and abiding by the essential requirements of "reasonable input control, reduction of energy consumption, improvement of green output, and reduction of pollution emission" in the land use process, a GLUUL efficiency evaluation index system was constructed, which includes input, desirable output, and undesirable output (Table 1). Its uniqueness lies in the comprehensive consideration of economic, social, and ecological output indicators to quantitatively analyze and maximize social, economic, and ecological output and minimize environmental loss in the urban land use course.

Input index: we select the labor force element, land element, and capital element as the input index. The total employment is chosen to represent the labor force element [48], the urban built-up area is selected to stand for the land element [49], and the capital element is represented by the total investment in urban fixed assets [48].

Desirable output: we select economic output, social output, and ecological output as indicators. Economic output is represented by the added value of the secondary and tertiary industries [11]. Social output is denoted by the average wage of urban workers [50]. Ecological output is defined as the total carbon sink of an urban green space [13], which is mainly a dynamic procedure of urban green vegetation absorbing $CO_2$ in the air through photosynthesis. The formula is as follows:

$$C_S = A_S \times f_S \qquad (1)$$

In Equation (1), $C_S$ is the total amount of $CO_2$ removed from the atmosphere by urban green vegetation, $A_S$ is the urban vegetation cover, and $f_S$ is the carbon sink coefficient. According to the relevant research [51,52], the value of $f_S$ is chosen to be 1.66.

**Table 1.** GLUUL efficiency evaluation index system.

| First-Order Index | Secondary Index | Measure Index | Unit | Type | References |
|---|---|---|---|---|---|
| Input | Labor force element | Total employment | $10^3$ people | + | Han et al. [48] |
| | Land element | Urban b uilt-up area | $km^2$ | + | Koroso et al. [49] |
| | Capital element | Total investment in urban fixed assets | CNY $10^4$ | + | Han et al. [48] |
| Desirable output | Economic output | The added value of the secondary and tertiary industries | CNY $10^4$ | + | Wang et al. [11] |
| | Social output | The average wage of urban workers | CNY | + | Xie et al. [50] |
| | Ecological output | Total carbon sink of the urban green space | $10^3$ tons | + | Tan et al. [13] |
| Undesirable output | Carbon emission | Carbon emissions of urban construction land | $10^3$ tons | - | Shan et al. [53] |
| | Environmental pollution emissions | Industrial pollution emissions | $10^3$ tons | - | Dong et al. [7] |

Undesirable output indicators: we mainly take carbon emissions and environmental pollution emissions as indicators. Carbon emission output is measured by urban construction land, which is calculated by the energy consumption of social production and living activities within the space range of urban built-up areas [53]. Urban carbon emissions come from both direct and indirect energy consumption, like natural gas and LPG, as well as electric energy and thermal energy consumption. The total of each city can be obtained by the sum of the carbon released from the consumption of natural gas, LPG, electric energy, and thermal energy [7]. Environmental pollution emissions are gauged by industrial pollution emissions, which include wastewater, $SO_2$, soot, and nitrogen oxide [48].

HED is a development mode, which states that "innovation becomes the primary engine, coordination becomes the endogenous feature, green becomes the universal form, openness becomes the necessary route, and sharing becomes the fundamental goal". By referring to the relevant literature [26,31], an evaluation index system of HED level was established, including five categories of indicators, namely, innovation, coordination, green, openness, and sharing (Table 2).

**Table 2.** HED level evaluation index system.

| First-Order Index | Secondary Index | Measure Index | Unit | Type |
|---|---|---|---|---|
| Innovation | Innovation input | R&D internal expenditure/GDP | % | + |
| | | R&D employees/total employees | % | + |
| | Innovation output | Patent applications per 10,000 people | piece | + |
| | | Patents granted per 10,000 people | piece | + |
| Coordination | Urban–rural coordination | Rural–urban disposable income ratio | % | - |
| | Industrial coordination | The value added of secondary and tertiary industries/GDP | % | - |
| | | The value added of tertiary industry/value added of secondary industry | % | + |
| | Regional coordination | Regional GDP per capita/national GDP per capita | % | + |
| Green | Environmental pollution | Industrial wastewater discharge per unit of GDP | ton | - |
| | | Industrial waste gas discharge per unit of GDP | ton | - |
| | Environmental protection | Domestic sewage treatment rate | % | + |
| | | Comprehensive utilization rate of industrial solid waste | % | + |
| Openness | Foreign trade | Total imports and exports of goods/GDP | % | + |
| | Utilization of foreign capital | Actual utilized foreign capital in that year/GDP | % | + |
| | | Number of foreign-invested enterprises | person | + |
| Sharing | Cultural and education | Number of college students per 10,000 people | people | + |
| | | Education expenditure/general public budget expenditure | % | + |
| | Medical and health care | Number of beds in hospitals and health centers per 100 people | sheet | + |
| | Social security | Urban social insurance participation rate | % | + |

*3.2. Research Methods*

3.2.1. Super-Efficient SBM Model Incorporating Unexpected Outputs

The difference and advantage of the super-efficiency SBM model from other efficiency measurement models is that its evaluation dimension includes non-expected output. It not only continues the performance of the SBM model but also has the advantages of the super-efficient DEM model. On this basis, it can also effectively deal with the relative

efficiency ranking problem of decision-making units and truly reflect the relative size of GLUUL efficiency [13]. The mathematical expression is as follows:

$$\rho^* = min \frac{1 + \frac{1}{m}\sum_{i=1}^{m} \frac{D_i^-}{x_{ih}}}{1 - \frac{1}{s_1+s_2}\left(\sum_{r=1}^{s_1} \frac{D_r^g}{y_{rh}^g} + \sum_{k=1}^{s_2} \frac{D_k^b}{y_{kh}^b}\right)} \tag{2}$$

$$s.t. \begin{cases} x_{ik} \geq \sum_{j=1,j\neq h}^{n} \lambda_j x_{ij} - D_i^-, \ i = 1,\ldots,m \\ y_{rh}^g \geq \sum_{j=1,j\neq h}^{n} \lambda_j y_{rj}^g + D_r^g, \ r = 1,\ldots,s_1 \\ y_{kh}^b \geq \sum_{j=1,j\neq h}^{n} \lambda_j y_{kj}^b - D_k^b, \ k = 1,\ldots,s_2 \\ 1 - \frac{1}{s_1+s_2}\left(\sum_{r=1}^{s_1} \frac{D_r^g}{y_{rh}^g} + \sum_{k=1}^{s_2} \frac{D_k^b}{y_{kh}^b}\right) > 0 \\ D^- \geq 0, D^g \geq 0, D^b \geq 0 \end{cases} \tag{3}$$

In Equations (2) and (3), $\rho^*$ is GLUUL, $n$ is the number of DMU, and each DMU consists of $m$ input, $s_1$ the expected output, and $s_2$ the unexpected output. $X \in R^m$, $y^g \in R^{s_1}$, and $y^b \in R^{s_2}$ are the input, expected output, and unexpected output vectors, respectively, and $\lambda$ represents the weight of the corresponding input or output element. The matrixes are $X = [x_1,\ldots,x_n]\in R^{m\times n}$, $Y^g = \left[y_1^g,\ldots,y_n^g\right]\in R^{s_1\times n}$, and $Y^b = \left[y_1^b,\ldots,y_n^b\right]\in R^{s_2\times n}$. $D^-$, $D^g$, and $D^b$ are the input, expected output, and unexpected output slack variables, respectively.

3.2.2. Entropy Method

To measure the level of HED through multiple dimensions, it is necessary to assign weights to indicators of different dimensions. The entropy method constructs the optimal weight by calculating the contribution of uncertain factors in the system. Compared with the weights determined by the expert review method, the coefficient of variation method, and the analytic hierarchy process, the entropy approach prevents human factors from interfering and provides a more comprehensive and rational expression of the utility value of information entropy. Therefore, the entropy method is used to evaluate the HED level in this paper. The mathematical expression is as follows.

The following equation is example 1:

$$D_{ijt} = \begin{cases} \frac{X_{ijt} - min(X_{ijt})}{max(X_{ijt}) - min(X_{ijt})} + d \\ \frac{max(X_{ijt}) - X_{ijt}}{max(X_{ijt}) - min(X_{ijt})} + d \end{cases} \tag{4}$$

$$E_j = -k\sum_{t=1}^{r} \sum_{i=1}^{m} P_{ijt}\ln(P_{ijt}) \tag{5}$$

$$W_j = \frac{1 - E_j}{\sum_{j=1}^{n}(1 - E_j)} \tag{6}$$

$$C_{it} = \sum_{j=1}^{n} W_j D_{ijt} \tag{7}$$

In Equations (4)–(7), $X_{ijt}$ stand for the $j$ index value of the region $i$ in the year of $t$. The $min(X_{ijt})$ and $max(X_{ijt})$, respectively, represent the minimum and maximum value of the index $j$. $D_{ijt}$ indicates the value of each index data after standardized processing. To ensure that the subsequent calculation is meaningful, the standardized matrix is translated by $d$

units. To reduce the impact of translation on data, the value of $d$ is $10^{-5}$. Then, $E_j$ is the entropy value of the item $j$, where, $k = 1/ln(r \times m)$, $P_{ijt} = D_{ijt}/\sum_{t=1}^{r} \sum_{i=1}^{m} D_{ijt}$, $W_j$ is the weight of the item $j$, and $C_{it}$ represents the HED level of the region $i$ in the year $t$.

### 3.2.3. Coupling Coordination Degree (CCD)

The coupling degree is used to measure the level of interaction and mutual influence between two or more systems in the development process frequently [54]. If the systems cooperate with each other and are complementary, it is benignant coupling. On the contrary, if the systems repel each other and are frictional, it is malignant coupling. Overall, more coupling means a stronger extent of interaction between systems and a stronger the coupling correlation, but it does not rule out the existence of the "unveracious high coupling" phenomenon when all the levels are low. Therefore, coordination degree should be further introduced to measure the size of benign coupling in the interaction between systems to reflect the overall coordination and consistency of the system. The CCD model of GLUUL and HED was constructed, and the mathematical expression is as follows:

$$C = \sqrt[2]{\eta_1 + \eta_2}/(\eta_1 + \eta_2) \tag{8}$$

$$T = \alpha\eta_1 + \beta\eta_2 \tag{9}$$

$$D = \sqrt{C \times T} \tag{10}$$

where $\eta_i (i = 1, 2)$ represents GLUUL efficiency and HED level, respectively. $C$ is the coupling degree of GLUUL and HED. $T$ is the comprehensive coordination index, and $\alpha$ and $\beta$ are undetermined coefficients. In the urbanization process, improving GLUUL is equally as important as promoting HED, so $\alpha = \beta = 0.5$ [55]. $D$ is the coupling coordination degree, and $0 \leq D \leq 1$. The closer $D$ is to 0, the worse the coupling and coordination development of GLUUL and HED system is. When $D$ is close to 1, it indicates that GLUUL and HED in this region are in a highly coordinated development level. According to present research results and actual conditions [56], the value of $D$ is divided into eight intervals (see Table 3), representing different coupling levels and coordination degrees.

**Table 3.** The criterion of the GLUUL and HED coupling coordination degree.

| Degree of Coordination | Coupling Coordination Degree | Coupling Level |
|---|---|---|
| Antagonistic range | $0.0 \leq D \leq 0.1$ | Severe disorder |
| | $0.1 < D \leq 0.3$ | Moderate disorder |
| | $0.3 < D \leq 0.4$ | Mild disorder |
| Run-in range | $0.4 < D \leq 0.5$ | Borderline disorder |
| | $0.5 < D \leq 0.6$ | Primary coordination |
| Coordinated range | $0.6 < D \leq 0.7$ | Intermediate coordination |
| | $0.7 < D \leq 0.9$ | Good coordination |
| | $0.9 < D \leq 1.0$ | Superior coordination |

After determining the CCD of each city, the regional development type can be further determined according to the ratio of $\eta_1$ and $\eta_2$ (*E*). Due to the factors of the two systems, which are in different dimensions, and the magnitude of the values being quite disparate, these data are not suitable to compare directly. Hence, this paper refers to the judgment criteria by Zhu [53] and makes improvements based on the actual situation, enlarging $\eta_2$ by 10 times and then comparing it with $\eta_1$. Since the level of GLUUL cannot be the same as HED in a city ($\eta_1/\eta_2 = 1$), the approximate value ($\eta_1/\eta_2 \approx 1$) is regarded as the synchronous development of the two systems. To minimize the error and improve comparability, $1 \pm 0.2$ was taken as the basis to judge whether the two systems were synchronized [57], and the development types of cities were divided into three types (see Table 4).

**Table 4.** Criterion of development types of cities.

| Type of Development | $E=\eta_1/\eta_2$ |
|---|---|
| Advanced development | $E < 0.8$ |
| Synchronous development | $0.8 \leq E \leq 1.2$ |
| Backward development | $E > 1.2$ |

3.2.4. Geographically and Temporally Weighted Regression (GTWR)

GTWR is an extension of the GWR in time dimension, which can simultaneously capture the temporal and spatial heterogeneity of variable data. It has been widely used in Geographic Information-Science, Environmental Science, Hydrology, Epidemiology, and other research fields [51,58,59]. Therefore, we used it to explore the social and economic factors influencing the coupling coordination of GLUUL and HED and to explore the spatial–temporal differentiation characteristics. The mathematical expression is as follows:

$$Y_i = \beta_0(\mu_i, v_i, t_i) + \sum_{k=1}^{K} \beta_k(\mu_i,\ v_i,\ t_i)X_{ik} + \varepsilon_i \tag{11}$$

where $Y_i$ is the value of the i-th explained variable and $(\mu_i, v_i, t_i)$ is the longitude, latitude, and time coordinates of the i-th sample point. $\beta_0(\mu_i, v_i, t_i)$ is the regression intercept and $\beta_k(\mu_i, v_i, t_i)$ is the regression coefficient of the k-th explanatory variable. $X_{it}$ is the data of the k-th explanatory variable and $\varepsilon_i$ is the error term. The selection of bandwidth will affect the model results. Too small of a bandwidth value will lead to over-fitting, while too large of a bandwidth value will include points that have little impact on the model, resulting in inaccurate results. In this paper, adaptive bandwidth is adopted, namely, the revised Akaike Information Criterion (AICc) is used as the criterion for bandwidth and model selection. The natural break point method in ArcGIS is used to divide the model results, and the intensity of the influencing factors is divided into five levels: strong, stronger, medium, weaker, and weak, which are displayed in different colors on the map. Blue is weak, green is weaker, yellow is medium, orange is stronger, and red is strong.

*3.3. Research Area and Data Description*

As the largest developing country in the world, China's urbanization process is ongoing and is currently at a critical stage of transitioning from resource- and factor-driven primary urbanization to innovation- and talent-driven secondary urbanization [60]. Therefore, China must balance the relationship between urban land use and economic development while considering resource and environmental constraints. Taking it as a research object to quantitatively examine the coordination relationship between GLUUL and HED can provide a valuable reference for other developing countries or regions. Due to the unavailability and inaccuracy of data on GLUUL efficiency-related measures at the municipal level, some cities with missing data were excluded from the study area. This paper identified 282 cities throughout China as the study object, including 33 cities in North China, 33 cities in Northeast China, 77 cities in East China, 42 cities in Central China, 36 cities in South China, 31 cities in Southwest China, and 30 cities in Northwest China. Their specific distribution is shown in Figure 2.

Data come from the "China Urban Statistical Yearbook", "China Environmental Statistical Yearbook", and "China Urban Construction Statistical Yearbook" over the past years.

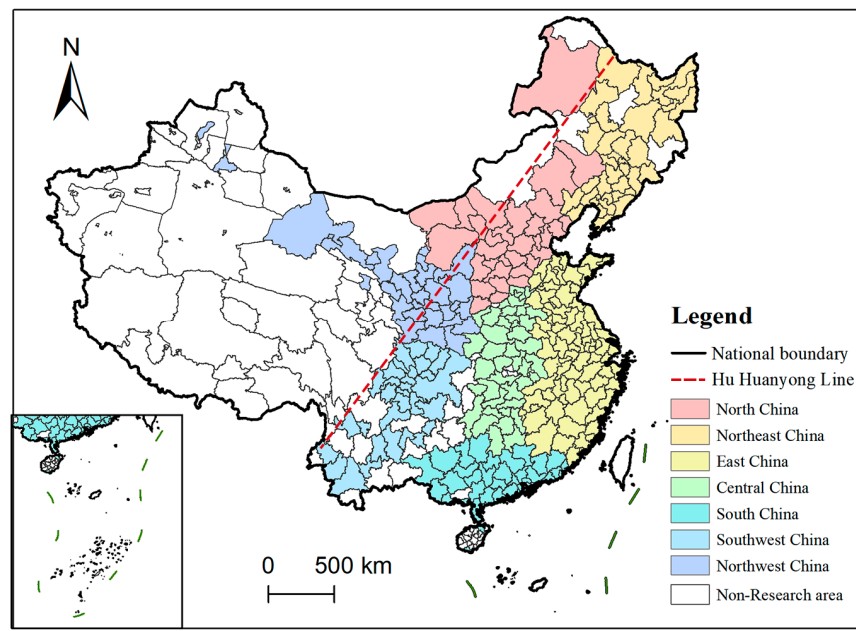

**Figure 2.** Research area.

## 4. Empirical Results

### 4.1. Analysis of GLUUL Efficiency and HED Level

Seven geographical administrative divisions were taken as the dividing standard: North China, Northeast China, East China, Central China, South China, Southwest China, and Northwest China. Their mean values were taken as the development levels of GLUUL and HED in different areas, and the results are presented in Figure 3.

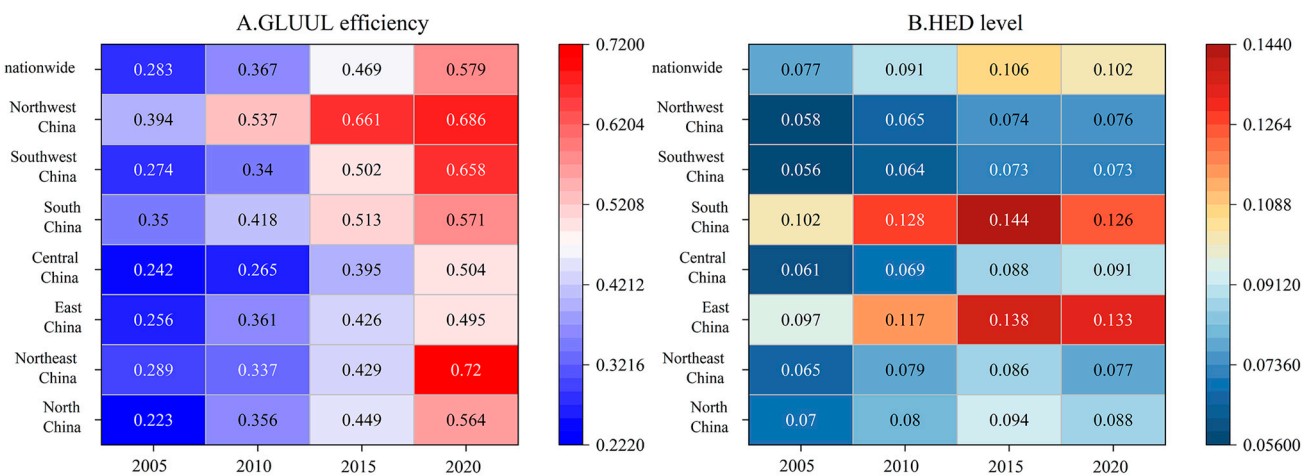

**Figure 3.** Time evolution characteristics of GLUUL efficiency and HED level. Panels A and B depict the change in GLUUL efficiency and HED level, respectively.

During the whole study period, GLUUL efficiency was significantly improved, showing a steady increase trend. The mean value rose from 0.283 in 2005 to 0.579 in 2020, with the largest increase in Northeast China followed by Southwest China, and the smallest increase in East China. With the promotion of ecological civilization construction and the implementation of a new urbanization strategy, the development mode of green transformation is accelerated [61]. GLUUL efficiency in Northeast China has been greatly improved under the substantial support of the "comprehensive revitalization" strategy [62].

HED level showed a fluctuant growth trend, which is characterized by an inverted "V" shape during 2010–2020. The average value rose from 0.077 in 2005 to 0.102 in 2020,

among which East China had the largest increase and Northeast China had the smallest increase. East China has a good industrialization foundation, an early start in economic expansion, and a strong foundation of a modern industrial system. In addition, the central government encourages the acceleration of development in the eastern region, greatly improving its HED level [63]. The industrial structure of Northeast China is relatively undiversified and over-relies on resource-based industries. However, in recent years, the lower demand for domestic downstream industries increased the downward pressure on steel, coal, petrochemical, and other industries [64]. As a result, Northeast China, which relies on these industries, has been greatly affected.

To show the evolution of GLUUL efficiency and HED level in different regions clearly and intuitively, the measured values were imported into ArcGIS 10.8 to draw the spatial–temporal evolution trends of four years in 2005, 2010, 2015, and 2020 (Figures 4 and 5). The efficiency of GLUUL was classified into five levels using the natural breakpoint method in the software. The levels are as follows: low level [0.0304, 0.1982), medium–low level [0.1982, 0.2943), medium level [0.2943, 0.4641), medium–high level [0.4641, 0.7837), and high level [0.7837, 1.4590). The HED levels were classified into five grades using the quantile method. The grades are as follows: low level [0.0209 to 0.0459), medium–low level [0.0459, 0.0555), medium level [0.0555, 0.0655), medium–high level [0.0655, 0.0982), and high level [0.0982, 0.7024).

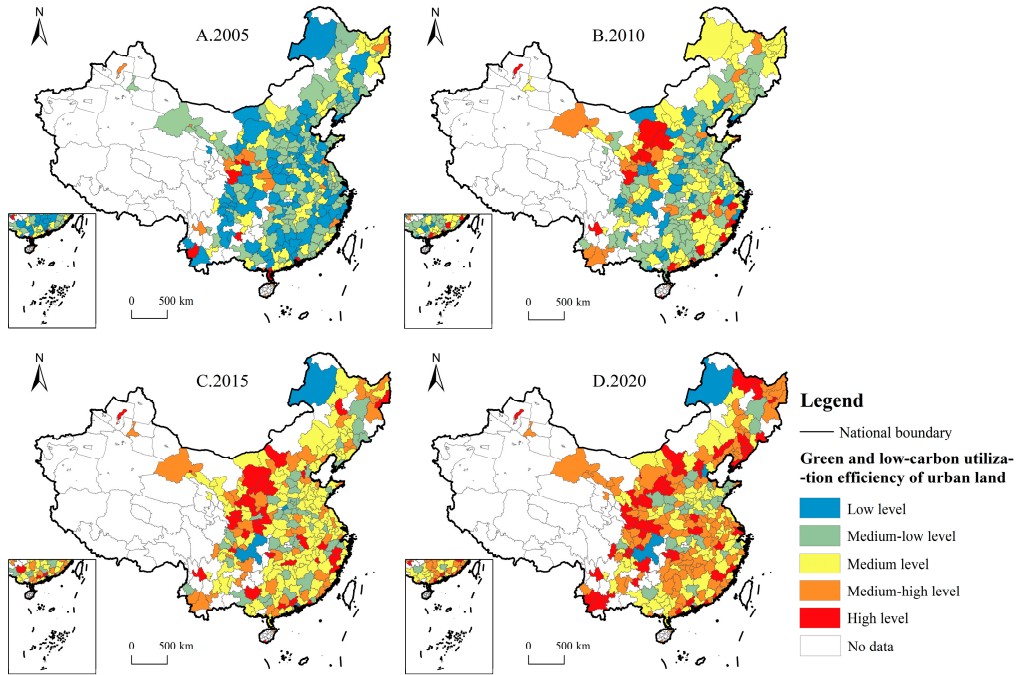

**Figure 4.** Spatial–temporal analysis of GLUUL efficiency. Panels (**A**–**D**) depict the changes in GLUUL efficiency in 2005, 2010, 2015, and 2020, respectively.

Figure 4 illustrates that in 2005, GLUUL efficiency was at a low or medium–low level in most regions, and a few cities with high or medium–high levels were mainly concentrated in Gansu Province. In 2010, the number of low-level cities decreased significantly. Most of them lay in the medium–low or medium level. The high-level and medium–high-level cities were obviously concentrated in the intersection of Gansu, Ningxia, and Inner Mongolia. In 2015, the number of medium-level cities increased drastically, lower-level cities only included Chongqing, Hulunbuir, Jiaozuo, and Dongguan. High and medium–high-level cities were concentrated along the "Hu Huanyong Line" and the southeast coast. In 2020, the number of cities with high and medium–high levels further increased, accounting for more than 60% of the total. Chongqing and Hulunbuir lay in the low level, and Tianjin dropped to the low level.

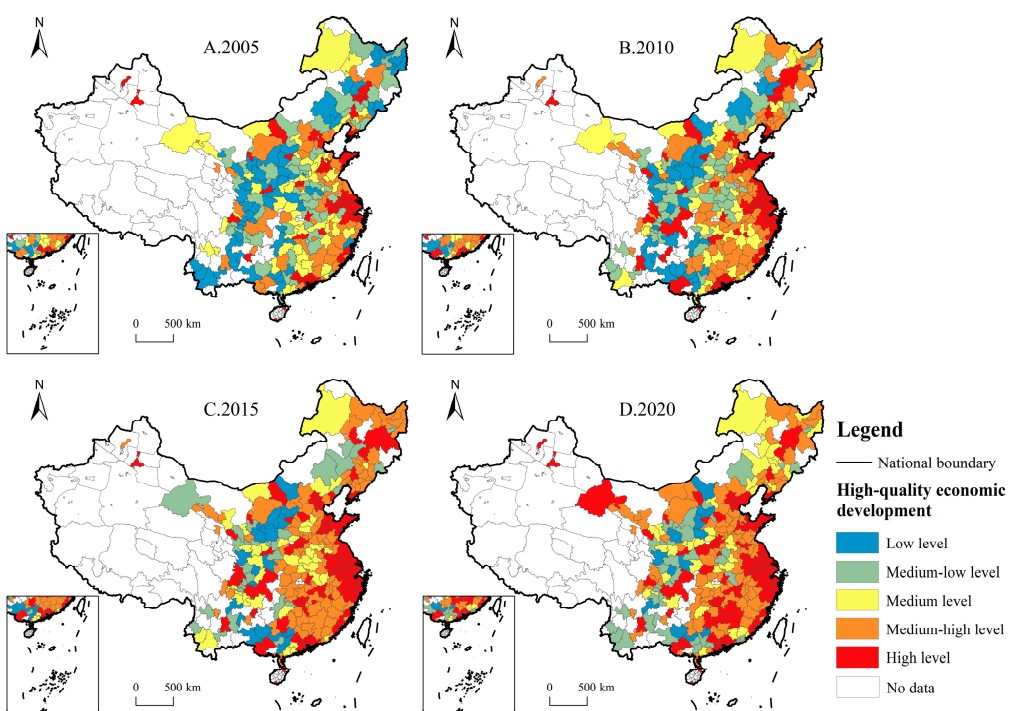

**Figure 5.** Spatial–temporal analysis of HED level. Panels (**A**–**D**) depict the changes in HED levels in 2005, 2010, 2015, and 2020, respectively.

Figure 5 shows that in 2005, cities with high and medium–high HED levels were predominantly located along the eastern coast, and the central region was mainly at medium and medium–low levels, while the western region was mainly at low levels. From 2010 to 2020, the distribution of HED levels decreased from east to west, which is essentially unchanged. The number of cities with high and medium–high levels of HED has risen significantly, reaching 69% in 2020, and most of them were in the east. There were only 12 low-level cities left, located dispersedly in the western region.

### 4.2. CCD Relationships of GLUUL and HED
4.2.1. Time Evolution Trend of CCD

Due to the number of cities studied being too much, it is inconvenient to display all the results, so they are presented in the form of charts. Figure 6 reflects the alterations in the number and proportion of different grades of cities. Overall, the main coupling level of cities evolved from primary coordination to intermediate coordination. The proportion of cities with good coordination gradually increased, while the proportion of cities with borderline disorder gradually decreased to near zero. In 2005, the main coupling level of the city was primary coordination, accounting for 67%, and the secondary was borderline disorder. The entirety was in the run-in range of development. In 2010, the major coupling level of the city was primary coordination, and the proportion increased slightly. The proportion of intermediate coordination exceeded the borderline disorder and became a secondary coupling level. In 2015, the main coupling level of the city was still primary coordination, but the proportion decreased, while the proportion of intermediate coordination increased. In 2020, intermediate coordination exceeded primary coordination and became the main coupling level of cities, accounting for 44%, while primary coordination decreased to 39%. In addition, cities with good coordination increased to 16%, and those three levels accounted for 99% of the total amount. This indicates that GLUUL and HED coupling coordination of China is about to enter a new stage and fully enter the coordination range.

Figure 7 shows the mean value of CCD between GLUUL and HED in different regions during 2005–2020. In the aspect of the time dimension, the average value of each region and the whole country presented a steadily increasing trend, with the overall mean value

rising from 0.5378 in 2005 to 0.6335 in 2020, exhibiting a growth rate of 17.8%. It indicates that GLUUL and HED were further harmonized in all regions during the investigation period. From the inter-regional comparison, in 2005, Northwest China had the highest mean value for CCD, followed by South China, and the lowest in North China. By 2020, the average value of CCD in Northeast China became the highest, followed by Northwest China, with the lowest value expected in Central China. Throughout the period of study, Southwest China was the fastest moving towards harmonization with an average growth rate of 7.7%, followed by Northeast China with an average growth rate of 6.72%, and South China with the slowest rate of 4.19%.

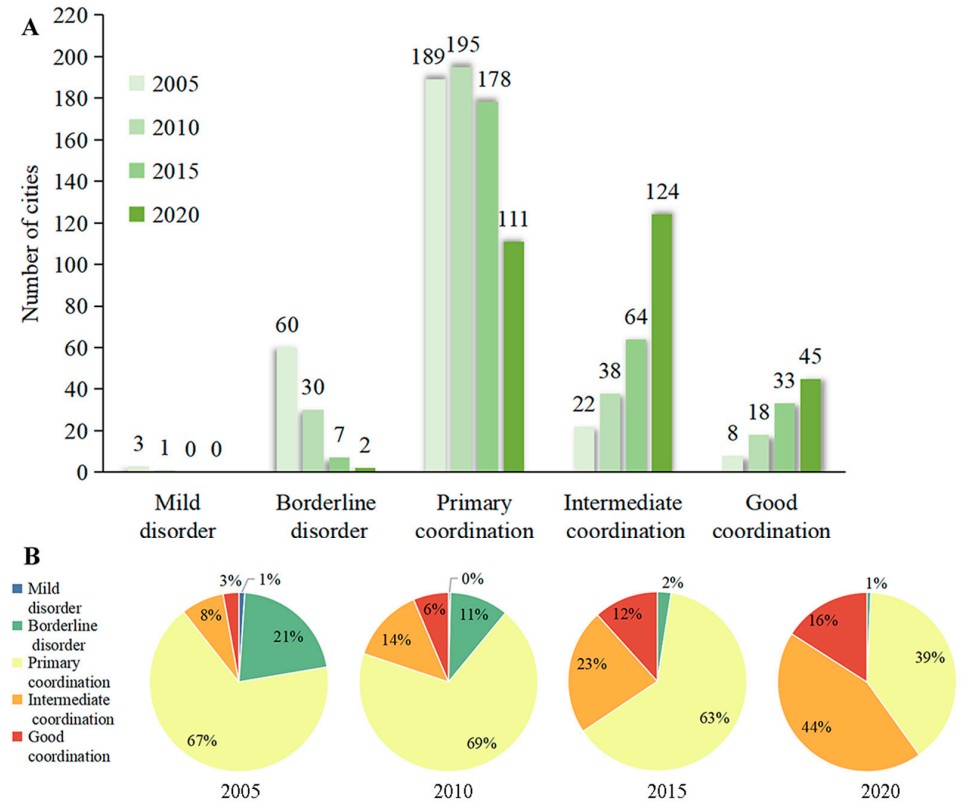

**Figure 6.** The changes in cities with different coupling levels. Panels (**A**,**B**) depict the changes in number and proportion, respectively.

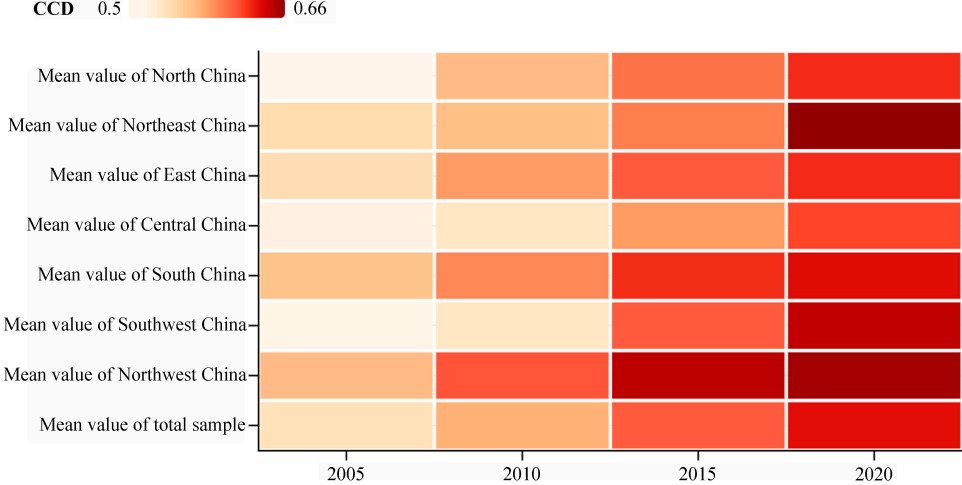

**Figure 7.** Regional and national CCD averages vary over time.

4.2.2. Spatial Evolution Trend of CCD

As demonstrated in Figure 8, the spatial distribution and evolution mode of CCD presented "the cities around Hu Huanyong Line and southeast coastal area radiate central region" generally. Specifically, in 2005, the top ten cities in CCD were Shenzhen, Zhanjiang, Dongguan, Haikou, and Sanya in South China, Anshun and Lincang in Southwest China, and Shangluo, Pingliang, and Longnan in Northwest China, which were in the stage of intermediate coordination and good coordination. Dongguan and Shenzhen, located in the southern coastal region, have taken the lead in achieving a high degree of coupling coordination. Their GLUUL efficiency and HED levels were among the highest in China. This is due to their advantages in technological innovation and tertiary development, which can be verified by basic index data (e.g., Dongguan's and Shenzhen's R&D internal expenditure is the first and second in China, respectively). The bottom ten cities were Chongqing, Neijiang, Liupanshui, Baoshan, Zhaotong, and Pu'er in the Southwest; Laibin in South China; Zhongwei in the Northwest; and Yuncheng and Xinzhou in North China. Among them, Liupanshui and Pu'er showed mild signs of disorder, while the remaining cities were on the verge of disorder. In 2010, the spatial distribution of CCD first emerged, with higher levels occurring in the northwestern and southeastern regions and lower levels occurring in the central region. The top ten cities in CCD were Jinhua, Quanzhou, and Yichun in East China, Dongguan and Sanya in South China, Lijiang in Southwest China, Jiayuguan, and Qingyang, Guyuan, and Karamay in Northwest China, which were all in the good coordination stage. The bottom ten cities were Chongqing, Meishan, and Liupanshui in the Southwest; Guigang, Hezhou, and Laibin in South China; Nanyang in Central China; Xinzhou in North China; Baoji in the Northwest; and Chaoyang in the Northeast. Among them, only Liupanshui lay in the mild disorder stage. In 2015, the cities with high CCD spread from Northwest China's Gansu Province to the Northeast and Southwest. Meanwhile, a "coastal line" of high CCD was formed in the southeast, and most cities entered the coordination stage. The top 10 cities were mainly distributed in southern China, including Guangzhou, Shenzhen, Foshan, Hechi, and Sanya, which were all in the good coordination stage. Cities at the bottom have progressed to the primary coordination stage, except for Chongqing, Laibin, and Hulunbuir, which were on the verge of disorder. From 2010 to 2015, Chongqing Municipality vigorously developed its automobile industry and transport infrastructure. Meanwhile, along with the disorderly expansion of its urban area, its GLUUL efficiency was suppressed and has remained consistently low (less than 0.1), which led to a lower CCD than that of most cities. In 2020, the number of cities with high CCD was aggrandized and distributed around the "Hu Huanyong Line" and southeast coastal areas, which has formed an envelope pattern and radiated cities in the central region. Currently, the top ten cities in CCD are Beijing in North China, Dalian and Tieling in Northeast China, Shanghai and Suzhou in East China, and Guangzhou, Shenzhen, Zhuhai, Foshan, and Chaozhou in South China, which are all in the good coordination stage. The CCD of some cities fell back, among which Jinhua, Changsha, and Hechi showed a significant decline, from good coordination to primary and intermediate coordination. The reason for this may be that the increase in energy consumption in these cities in 2020 led to a significant rise in carbon emissions, which reduced the efficiency of GLUUL, and, consequently, the coupling coordination. The cities at the bottom of the list were Yuncheng, Linfen, and Hulunbeier in North China; Jilin in the Northeast; Binzhou in East China; Shangqiu and Xiaogan in Central China; Shaoguan and Hechi in South China; and Chongqing in the Southwest.

4.2.3. Subtype Analysis of CCD

The purpose of studying CCD subtypes is to further understand which sub-system lagging behind in the coupled coordination relationship between GLUUL and HED at the present stage. This will help us make policy recommendations accordingly, so the most recent year in our research period, 2020, was selected to study. Cities with different D and

E were divided into different coupling coordination subtypes; the results were presented in the form of a map and table, as shown in Figure 9 and Table 5.

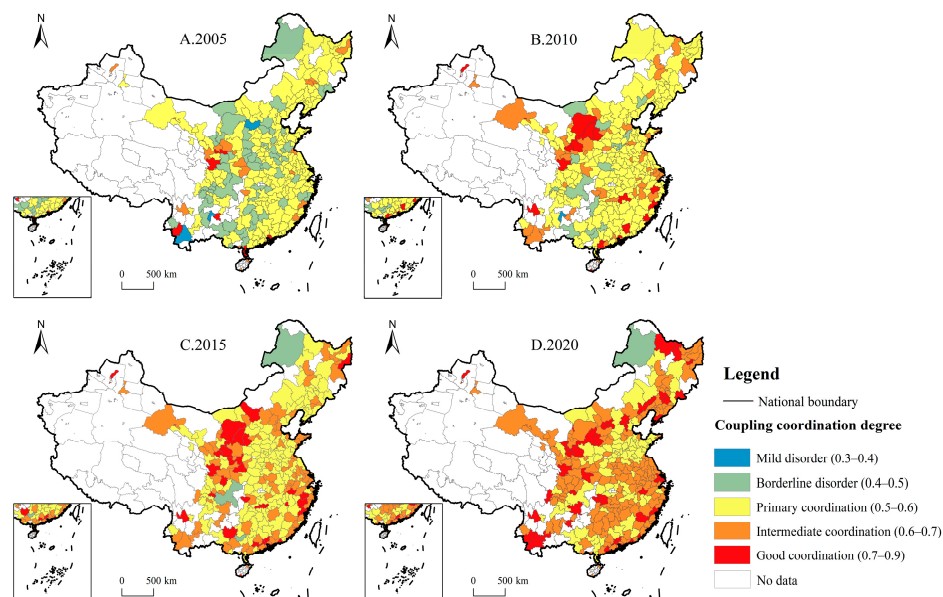

**Figure 8.** Spatial–temporal analysis of CCD. Panels (**A**–**D**) depict the changes in CCD in 2005, 2010, 2015, and 2020, respectively.

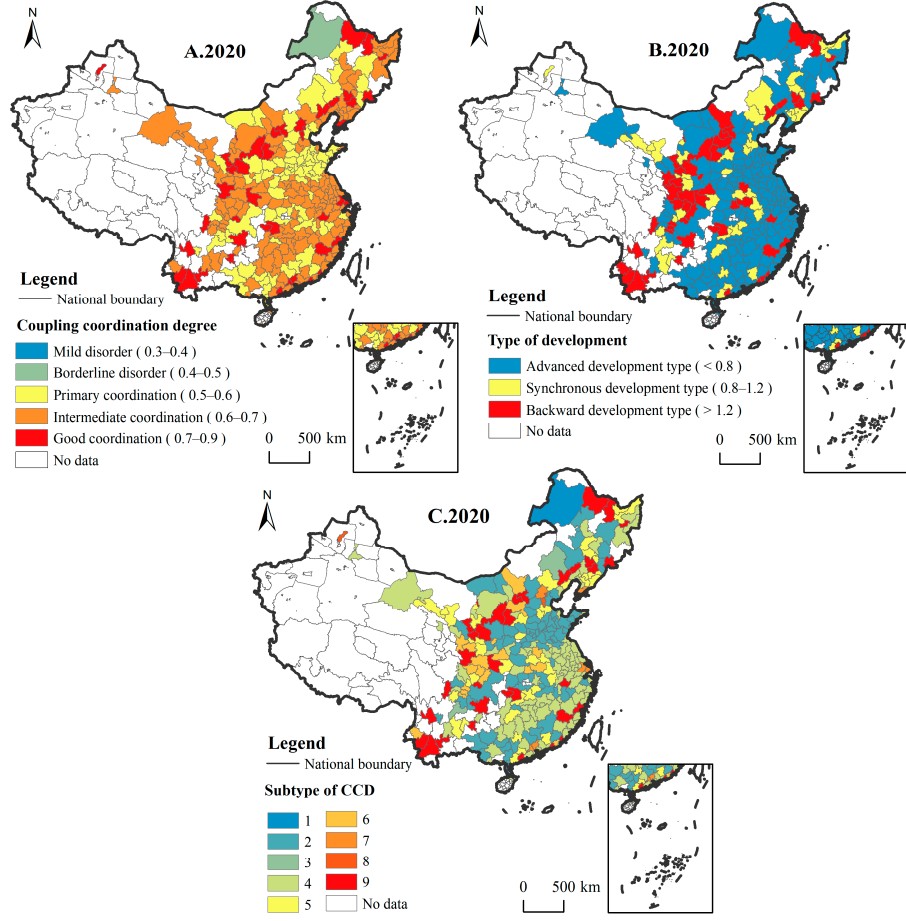

**Figure 9.** Spatial distribution of the subtypes of CCD. Panels (**A**–**C**) depict the distribution of CCD, type of development, and subtype of CCD in 2020, respectively.

**Table 5.** Number and proportion of subtypes of CCD (2020).

| Degree of Coordination | Number of Cities | CCD | Type of Development | Subtype of CCD | Number of Cities | Proportion |
|---|---|---|---|---|---|---|
| Run-in range | 79 | Borderline disorder | Advanced development type | 1 | 1 | 0.35% |
| | | Primary coordination | Advanced development type | 2 | 76 | 26.95% |
| | | | Synchronous development type | 3 | 2 | 0.71% |
| Coordination range | 203 | Intermediate coordination | Advanced development type | 4 | 98 | 34.75% |
| | | | Synchronous development type | 5 | 41 | 14.54% |
| | | | Backward development | 6 | 19 | 6.74% |
| | | Good coordination | Advanced development type | 7 | 9 | 3.19% |
| | | | Synchronous development type | 8 | 2 | 0.71% |
| | | | Backward development | 9 | 34 | 12.06% |

As can be seen in Figure 9, overall, GLUUL and HED coupling in 282 cities in China were basically coordinated, but there was still a big gap between them and superior coordination. Most of the cities with high CCD were backward and surrounded the "Hu Huanyong Line". The cities had primarily advanced development and were centralized in the southeast coastal area intensively. The number of synchronous developing cities was the least. They were evenly distributed in the Northeast and central regions.

According to Table 5 and Figure 9, there were 98 cities belonging to the "intermediate coordination—advanced development" type, accounting for 34.75% of the total sample. Most of them lay in East China, such as Fuzhou and Wenzhou. A few of them are located in Northeast China like Shenyang, and South China, like Sanya. These cities have entered the intermediate coordination stage, but GLUUL was relatively behind HED. Advances in science and technology should be strengthened to promote GLUUL, and the protection of the urban ecological environment should be put in a more highlighted position. There were 76 cities belonging to the "primary coordination—advanced development" type, comprising 26.95% of the entire sample. They were mostly found at the junction of North China, East China, and Central China, including Tianjin, Qingdao, Jinan, Luoyang, Shijiazhuang, etc. In these cities, the HED was relatively ahead of GLUUL. First, the economic progress of these urban areas has been rapid under political support. Second, the agglomeration of population and industry increases the demand for land use, and at the same time, the input may not be properly controlled. This results in high consumption of energy and emission of pollutants, which restricts the coupling and coordinated development. A total of 41 cities belonged to the "intermediate coordinated—synchronous development" type, accounting for 14.54% of the total sample. They were scattered in almost every region, except East China, such as Chengde and Xianning in the western region and Yueyang and Anqing in the central region. HED and GLUUL of these cities have basically achieved coordinated development. While moving towards a higher level of HED in the future, land resources should be used rationally to continuously promote GLUUL. Then, 34 cities belonged to the "good coordination—backward development" type, accounting for 12.06% of the total sample. They were mainly distributed near the "Hu Huanyong line", containing Ya'an, Guyuan, Chaoyang, Lincang, etc. These cities have entered the stage of good coordination, but HED was relatively behind GLUUL. In the future, based on maintaining the advantages of GLUUL, HED is supposed to be further accelerated.

## 5. The Main Influencing Factors—From the GTWR Perspective

The CCD of GLUUL and HED was influenced by multiple factors. Based on the coupling coordination mechanism, following the comprehensive and scientific principles, we selected economic development level (Edl), green technology innovation ability (Gtia), industrial structure upgrading (Isu), foreign investment scale (Fis), urban population size (Ups), and financial support (Fs) as the main influencing factors.

The Edl is inextricably linked to GLUUL efficiency. As the urban economy continues to grow, high density, multi-level, multi-angle, and multi-quality characterized the development and utilization of land resources, and the efficiency of GLUUL increased accordingly. Edl is measured by GDP per capita. Gtia can promote GLUUL. The application of green technologies has the potential to lower resource and energy consumption, minimize waste generation and emissions, and thus advance the efficiency of land use. Gtia is calculated by the number of green patent applications per capita. Isu helps optimize land use efficiency. In the industrial structure upgrading process, some outdated traditional industries will be eliminated, while new industries will gradually emerge. Emerging industries usually adopt more advanced technology and management methods, which can promote GLUUL. Isu is calculated by the proportion of added value generated by the tertiary sector compared to that of the secondary sector. Foreign investment generally has high environmental protection and sustainable development requirements, which are conducive to driving local governments and enterprises to improve GLUUL efficiency, thereby achieving energy conservation, emission reduction, and resource recycling. Fis is measured by the total value of industrial output of foreign-invested firms. Population agglomeration is a key driver of urban development, which can promote the centralized use of resources, thereby creating scale economies and improving the efficiency of GLUUL. Urban population is selected to measure Ups. By investing financially in science and technology, local governments support the research, development, and promotion of green and low-carbon technologies and products, attracting high-tech enterprises to locate in these areas, resulting in more efficient use of land and reducing environmental losses, which benefits economic development and GLUUL efficiency. Fs selects investment in science and technology to measure, as illustrated in Table 6.

**Table 6.** Influencing factors and measurement indicators of CCD.

| Influencing Factor | Measurement Index | References |
|:---:|:---:|:---:|
| Edl | GDP per capita | Huang et al. [52] |
| Gtia | Number of green patent applications per capita | Xu et al. [65] |
| Isu | Proportion of the tertiary industry's added value in secondary industriess | Gao et al. [20] |
| Fis | Total industrial output value of foreign-invested enterprises | Wan et al. [66] |
| Ups | Urban population | Kong et al. [67] |
| Fs | Investment in science and technology | Lu et al. [68] |

### 5.1. Spatial Autocorrelation Features

ArcGIS 10.8 was utilized to acquire the global Moran's I index of CCD (Table 7). The global Moran's I value and Z-value of CCD in 2005, 2010, 2015, and 2020 all passed the spatial autocorrelation test at a significance level of 1%. A global Moran's I index > 0 illustrates that CCD is positively correlated in general. From 2005 to 2020, the value of the global Moran's I and Z showed a fluctuating upward trend, indicating that the spatial agglomeration effect of CCD was enhanced.

**Table 7.** Global spatial autocorrelation of GLUUL and HED coupling coordination degree.

| Year | Moran's I | Z-Value | *p*-Value |
|------|-----------|---------|-----------|
| 2005 | 0.1059 | 5.2714 | 0.0000 |
| 2010 | 0.1760 | 8.6291 | 0.0000 |
| 2015 | 0.1254 | 5.1114 | 0.0000 |
| 2020 | 0.1492 | 6.0511 | 0.0000 |

*5.2. GTWR Data Verification*

Factors affecting CCD were tested by the GTWR regression model (Table 8). In the aspect of AICc, the value of adaptive bandwidth was −3386.77, which was lower than that of the fixed bandwidth. The model's effect improves as the value decreases. For the goodness of fit aspect, the $R^2$ and adjusted $R^2$ were both higher than 0.47, indicating that the model fits well.

**Table 8.** GTWR estimation results of influencing factors.

| Model Parameters | Sigma | Residual Squares | AICc | $R^2$ | Adjusted $R^2$ | Spatiotemporal Distance Ratio |
|------------------|-------|------------------|------|-------|----------------|-------------------------------|
| Value (Fixed) | 0.0508 | 2.9144 | −3351.88 | 0.4677 | 0.4649 | 3.2493 |
| Value (Adaptive) | 0.0504 | 2.8689 | −3386.77 | 0.4760 | 0.4732 | 0.2688 |

*5.3. Temporal and Spatial Differences of Influencing Factors*

Figures 10–15 show the effect of each of these factors on CCD.

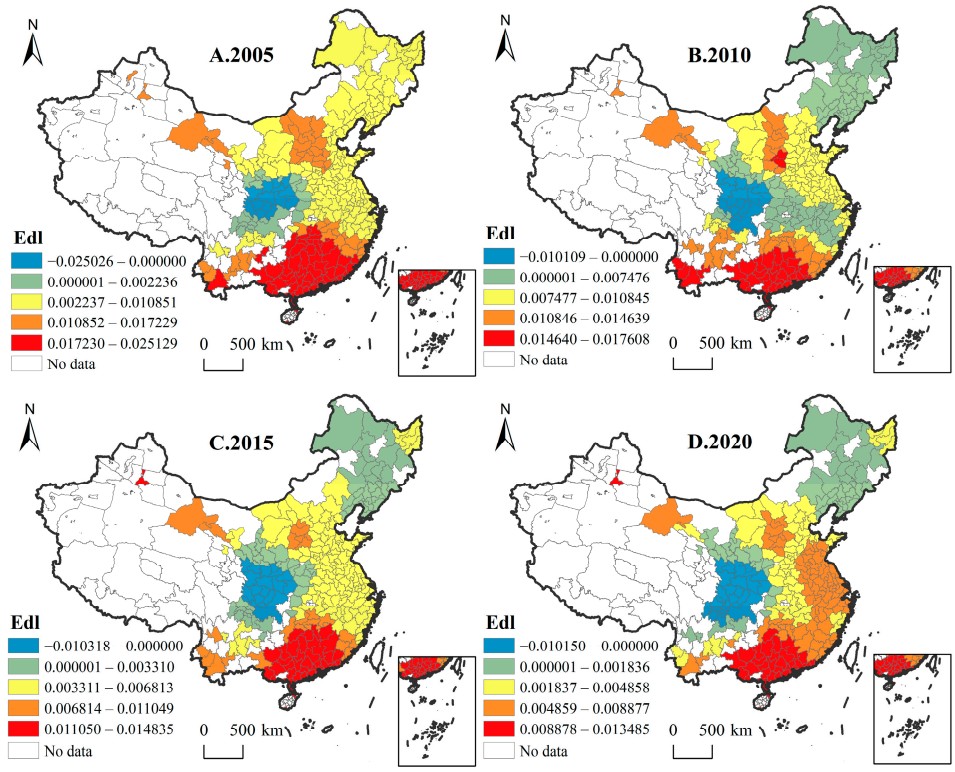

**Figure 10.** Spatial–temporal differentiation characteristics of Edl. Panels (**A**–**D**) depict the changes in Edl's influence in 2005, 2010, 2015, and 2020, respectively.

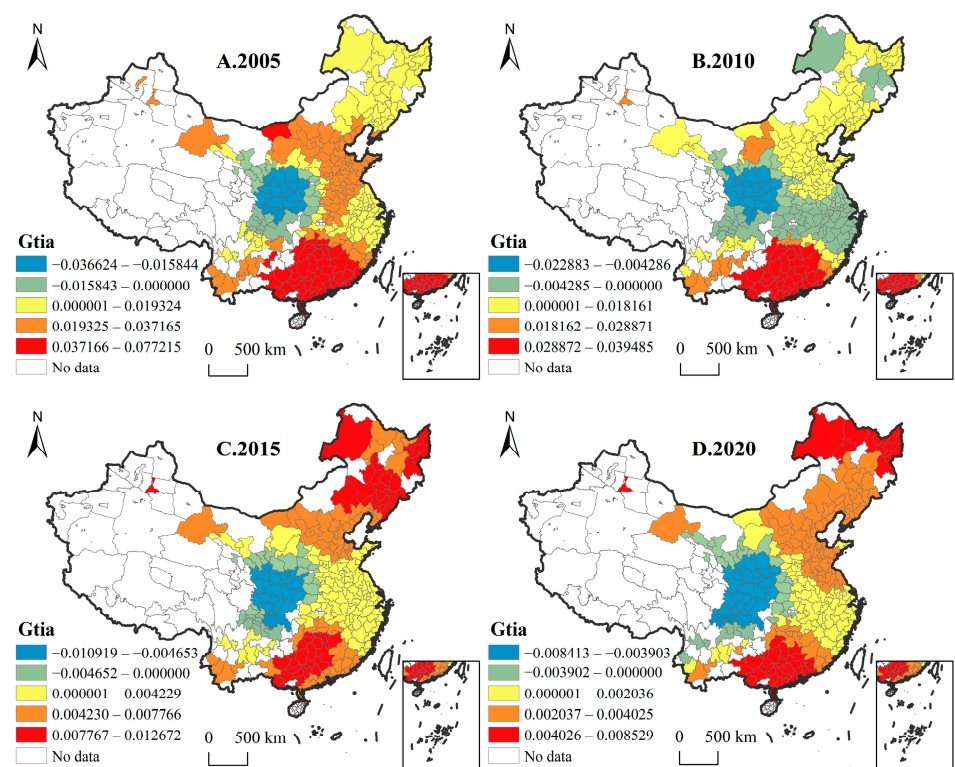

**Figure 11.** Spatial–temporal differentiation characteristics of Gtia. Panels (**A**–**D**) depict the changes in Gtia's influence in 2005, 2010, 2015, and 2020, respectively.

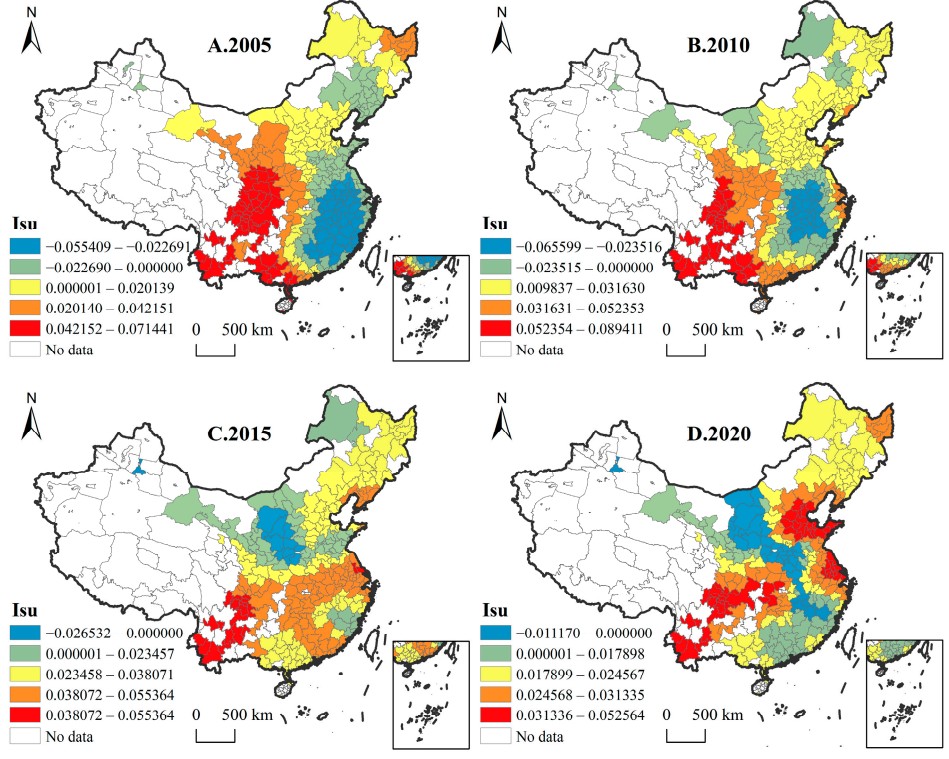

**Figure 12.** Spatial–temporal differentiation characteristics of Isu. Panels (**A**–**D**) depict the changes in Isu's influence in 2005, 2010, 2015, and 2020, respectively.

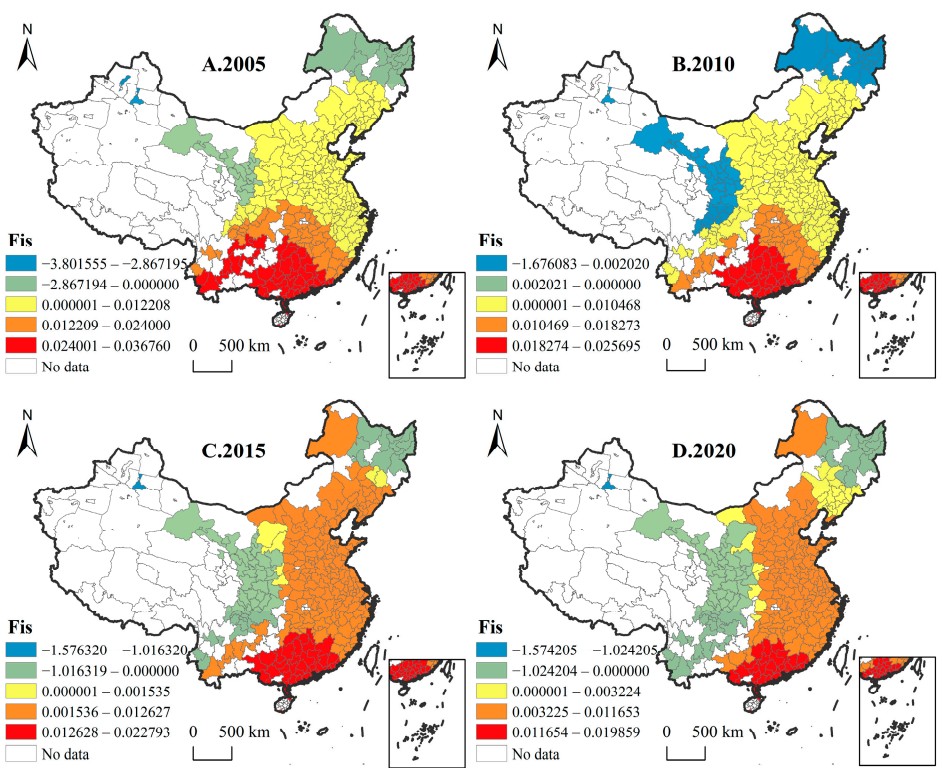

**Figure 13.** Spatial–temporal differentiation characteristics of Fis. Panels (**A**–**D**) depict the changes in Fis's influence in 2005, 2010, 2015, and 2020, respectively.

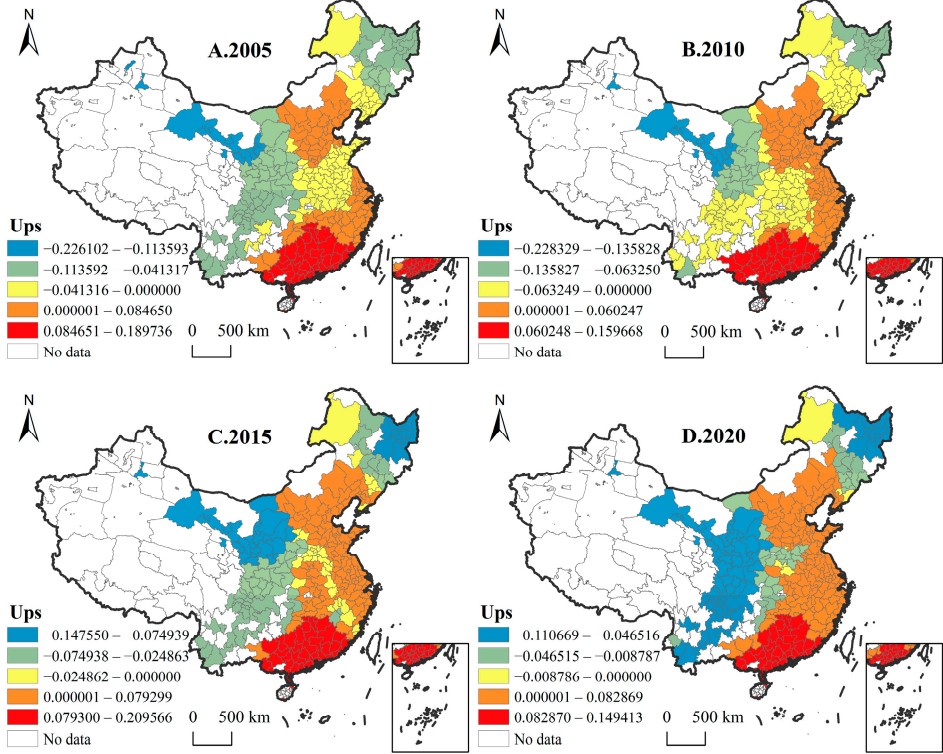

**Figure 14.** Spatial–temporal differentiation characteristics of Ups. Panels (**A**–**D**) depict the changes in Ups's influence in 2005, 2010, 2015, and 2020, respectively.

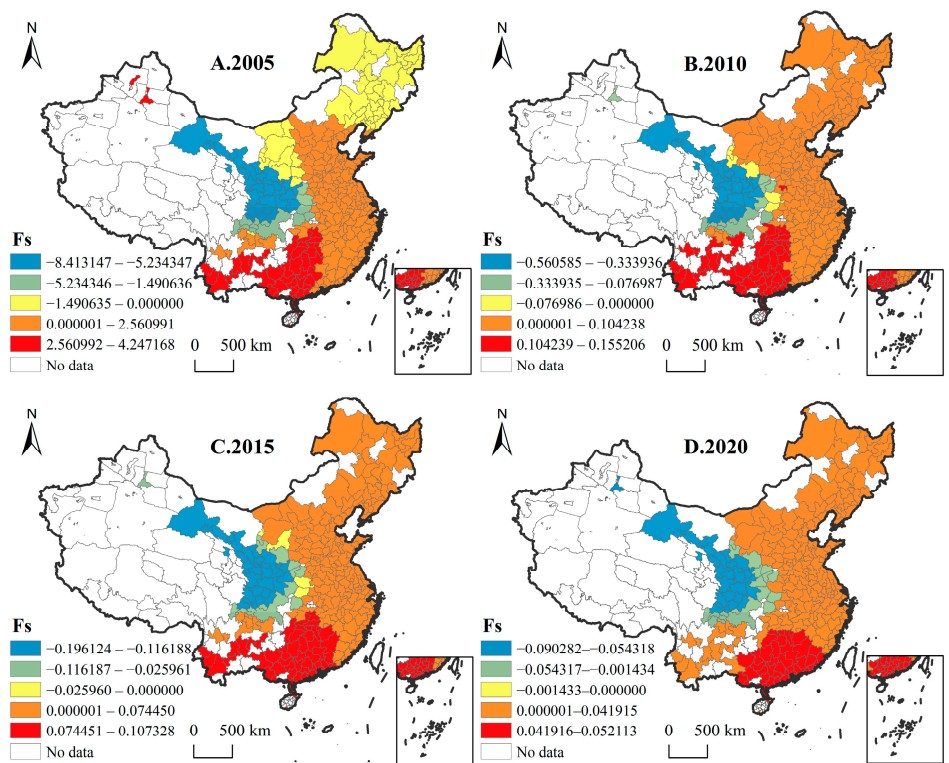

**Figure 15.** Spatial–temporal differentiation characteristics of Fs. Panels (**A**–**D**) depict the changes in Fs's influence in 2005, 2010, 2015, and 2020, respectively.

### 5.3.1. Economic Development Level

The influence of Edl on CCD is mainly promoting, and as its influence gradually decreases, the spatial–temporal non-stationarity weakens. This indicated that the improvement of Edl could promote CCD. As shown in Figure 10, according to the temporal and spatial differentiation of influence, Edl has the strongest positive promoting effect on Guangdong and Guangxi Province in South China, while it has a certain hindering effect on Chongqing and Sichuan in Southwest China, Shaanxi and Gansu Province in Northwest China. The overall influence pattern is relatively stable. From 2005 to 2020, the promoting effect of Edl on cities in East China was relatively enhanced, cities in Northeast China were relatively weakened. The cause might be that the economic progress of the southern region commenced earlier, subsequently experiencing consistent and rapid development after the reform and opening. Cities in East China, including Shanghai, Hangzhou, and Fuzhou, have followed suit. The economies in these regions are relatively developed, consequently exerting a strong promoting effect on CCD. Conversely, the western region's economic development is relatively deficient, and the natural land conditions are severe. Land desertification, salinization, and other problems lead to scarce available land resources. Meanwhile, with the industrial gradient transfer, resource and labor-intensive industries intensify the extensive use of land, further increasing the environmental pressure in the western region, which is not conducive to CCD. According to the variation process of the regression coefficient, the mean value decreases from 0.0098 to 0.0038. It shows that with the deepening of the development process, the limitation of promoting CCD solely by economic development gradually appears. The gap between the maximum and minimum decreased from 0.0502 to 0.0236. The regional unbalance of Edl's influence on CCD gradually decreases.

### 5.3.2. Green Technology Innovation Ability

Gtia has a beneficial impact on most cities in China; when its impact gradually weakens, the spatial–temporal non-stationarity decreases. This suggests that improving Gtia

may enhance CCD. From the temporal and spatial differentiation of influence (Figure 11), in 2005, Gtia had the strongest promotion effect on Guangdong, Guangxi, and Hunan Province in South China, while it had a certain inhibitory effect on cities in the western region. In 2020, the impact on cities in Shandong was relatively heightened, with the degree of impact on par with cities in the southern region. The pattern of influence distribution is relatively stable. The higher level of technological innovation in the eastern and southern regions may explain the overall differences observed with the central and western regions. Their technological innovation started earlier, and relevant policies were more precise. Reflecting on the quantity of applications for green patents, the top cities, including Beijing, Shanghai, Shenzhen, Hangzhou, Nanjing, Jinan, Tianjin, Shenyang, Dalian, etc., are predominately located in the east and south, while the central and western regions are generally below average. At the same time, these regions will undertake technology transfer from developed regions. New technologies and new industries may bring new land requisition and increase the bearing pressure of land. This promotes the formation of the affected pattern of "strong in southeast, weak in central and west". In addition, since the strategy of revitalizing Northeast China was launched in 2003, the northeast region has adhered to structural adjustment as the main line, introduced strategic emerging industries, and encouraged technological innovation. Green technology innovation in Northeast China can promote talent and industry agglomeration based on original land use, thus promoting CCD. From the change in the regression coefficient, the mean value gradually decreased from 0.0198 to 0.0013. The gap between the maximum and minimum decreased from 0.1138 to 0.0169. The regional unbalance of Gtia's influence on CCD narrowed over time.

### 5.3.3. Industrial Structure Upgrading

The overall influence of Isu on CCD is positive, and its influence gradually increases, and spatial–temporal non-stationarity is alleviated. This indicates that Isu can promote CCD. As the industrial structure continues to improve, its promoting effect is continuously enhanced. As can be seen in Figure 12, the influence of Isu has a significant temporal and spatial heterogeneity. In 2005, Isu had the strongest positive promoting effect on Southwest China, while it had a certain hindering effect on East China. The reason may be that the industrial structure upgrading index and HED level in East China were both high in the early phase, and industrial structure upgrading can promote regional economic development. Meanwhile, the development mode that emphasized speed and inefficiency also brings about high carbon emissions. As GLUUL efficiency was suboptimal, the pace of progress in both the land and economy systems did not correspond. Therefore, in this period, Isu hindered CCD in East China. From 2010 to 2020, the scope of the positive impact area was expanded. It has promoted the improvement of CCD significantly, except in Yuncheng, Linfen, and Luliang of Shanxi, Nanchang and Jiujiang of Jiangxi, and Yulin of Shaanxi. Among them, the Yangtze economic corridor in East China and the Beijing–Tianjin–Hebei region in North China are the most impacted. It is possible that the dominance of the eastern region in the development of tertiary industry during the later period of its evolution explains this phenomenon. Profiting from sufficient talent supply and a huge demand market, the eastern region has witnessed rapid growth in knowledge-intensive emerging sectors, including finance, R&D, and information services, reducing its dependence on energy-intensive and heavy industry for economic growth. This promotes GLUUL and narrows the development speed difference between land and economy, and thus promotes their coupling and coordination. From the perspective of the regression coefficient's change, the mean value grew from 0.0089 to 0.0212, and the gap between the maximum and minimum dropped from 0.1268 to 0.1113. The regional unbalance of Isu's influence on CCD decreased slightly over time.

### 5.3.4. Foreign Investment Scale

As shown in Figure 13, the influence of Fis on CCD has a significant spatial–temporal differentiation, and the overall influence trend has a great change. In 2005, Fis had a positive

impact on the eastern and southern regions, with a stronger influence in South China. The inhibitory effect was mainly concentrated in Xinjiang, Gansu, and Heilongjiang Provinces, and it was strongest in Xinjiang, located in Northwest China. From 2015 to 2020, the positive effect of Fis on South China was always the most significant, while the promoting effect on the central and eastern regions was relatively enhanced. The area of inhibition was expanded and concentrated in the western region and Northeast China. Xinjiang and Gansu Provinces in Northwest China and Heilongjiang Province in Northeast China were negatively correlated with Fis, but the degree of inhibition was gradually weakened. It is possible that the concentration of foreign investment in China was largely centered on the southern coastal areas, with the Pearl River Delta as its core, during the initial period of reform and opening, representing approximately 82% of the total, while the western region barely held a share of around 3%. The declining share of foreign investment in the southern region has been steadily shifting to the Yangtze River Delta and Beijing–Tianjin–Hebei urban clusters in the eastern region as reform and opening deepens. In 2020, the eastern region accounted for 85% of the total amount of foreign investment utilized, while the western region still only accounted for approximately 5%. When a region has a low level of foreign investment, it tends to increase carbon emissions. Because there was a strong drive towards rapid economic development, investment, and construction of factories regardless of the cost in the early stage of foreign capital introduction, the energy demand is large, and the effect of investment in environment-green on carbon emission reduction cannot be immediately shown, which hindered progress towards GLUUL. This impedes the enhancement of CCD in the western region. From the variation of the regression coefficient, the gap between the maximum and minimum decreased from 3.8383 to 1.5941. The regional unbalance of Fis's influence on CCD weakened over time.

### 5.3.5. Urban Population Size

The overall impact of Ups is positive, and its influence waned and then grew, while spatial–temporal non-stationarity slowed down. This indicates that Ups can promote CCD, and its promoting effect presents an inverted U-shaped pattern with the gradual growth of Ups. From the temporal and spatial differentiation of influence (Figure 14), in 2005, Ups promoted southeast coastal areas and central North China, and it promoted Guangdong, Guangxi, and Hunan Provinces in South China most significantly. It has a certain inhibitory effect on western regions and Northeast China, and Gansu Province in Northwest China is the most obvious. From 2010 to 2020, the area of positive impact continues to expand in the east. In 2020, Ups played a promoting role in East China, Northeast China, Central China, and South China, and the strongest positive impact was still concentrated in the southern coastal area. The inhibitory effect was more apparent in the western and northeastern regions. This may be due to the southeast coastal areas and the North China area having more favorable development environments, attracting population flow and concentration to them. The increase of population size and agglomeration of population is conducive to the improvement of building density, thus improving land use efficiency. Meanwhile, the population concentrations are also connected to the centralized energy supply, thus improving energy efficiency, which helps to promote GLUUL. However, the development of the western region lags behind other regions, resulting in substandard living and working conditions. The urban resources cannot match the growing population in a short time, resulting in the rapid increase of per capita energy consumption and environmental governance pressure. Since the implementation of the reform and opening policy, the northeastern region has experienced a loss of its geographical advantages, a significant decline in the regional economy, a heavy reliance on industrial sectors, and a continuous outflow of population. Population shrinkage restricts the economic density and energy supply level of northeast China and hinders its economic development. From the variation of the regression coefficient, the mean regression coefficient of Ups increased from 0.0023 to 0.0115. It indicates that the agglomeration of the urban population promoted the accumulation of resources, generated a scale economy, and positively acted on CCD. The

difference between the highest and lowest regression coefficients decreased from 0.4158 to 0.2601. The unbalanced regional impact of Ups on CCD was gradually mitigated.

5.3.6. Financial Support

As can be seen in Figure 15, the influence of Fs on CCD has a significant spatial–temporal differentiation, and the overall trend of influence is relatively stable. In 2005, Fs positively affected East China, Central China, South China, and central North China, but hindered Northeast China, Southwest China, and Northwest China to varying degrees. Among them, South China was the most affected by the positive effect, and Gansu Province in Northwest China was the most affected by the obstruction. From 2010 to 2020, the effect of Fs on Northeast China and North China all turned positive, while the pattern of influence in other regions is basically unchanged and gradually weakened from east to west. One of the possible reasons is the structural disparity in local financial spending on science and technology. The expenditure on science and technology in the eastern region greatly surpasses that of the central and western regions, with the central region having slightly more expenditures than the western region. Over time, there has been a growing rift between the three regions. An upsurge in government funding for science and technology is expected to drive economic expansion. It will bring technological innovation and make the transfer to green, low-carbon, and environmentally friendly energy, which has a favorable impact on both GLUUL and HED, thus promoting their coupling and coordination. In addition, the obstructive effect of Fs on the western region may arise due to various issues, such as an irrational financial allocation towards science and technology, inadequate funding for fundamental research, and low productivity of fiscal investment in science and technology in the western region. The mean regression coefficient of Fs increased from −0.1844 to 0.0186. It verified that the strengthening of financial support had a positive effect on CCD. The difference between the highest and lowest regression coefficients reduced from 12.6603 to 0.1424. There was a significant decrease in the disparity of the impact of Fs on CCD across different regions.

**6. Conclusions**

Based on an analysis of the coupling coordination mechanism, research was conducted on 282 cities in China. Comprehensive measurements and analysis were performed on the spatial–temporal characteristics of GLUUL and HED, as well as their CCD, from 2005 to 2020. Additionally, GTWR was used to determine the influencing factors of CCD and their action modes.

Firstly, during their development, GLUUL efficiency gradually improved, but the improvement extent reduced slowly. The HED system exhibited a fluctuating growth trend during 2010–2020, which is characterized by an inverted "V" shape. Regarding their spatial distribution, high-efficiency areas of GLUUL were mainly concentrated along the "Hu Huanyong Line". The regions manifesting high levels of HED are primarily located in the east, exhibiting a typical "high in the east and low in the west" trend, with a horizontal diffusion flowing towards the west.

Secondly, the spatial–temporal evolution of CCD indicates a growing overall harmony between 2005 and 2020. The main coupling level has progressed from primary coordination to intermediate coordination, resulting in a gradual increase in the proportion of cities exhibiting good coordination. Several cities teetered on the brink of chaos, and a pattern of relapse was evident. The distribution of CCD exhibited a relatively balanced and displayed declining spatial discrepancy. The general pattern of spatial distribution is characterized by being "high in the northwest and southeast, low in central", and the evolution trend is observed as "extending from the northwest to the southwest and northeast".

Thirdly, in 2020, 34.75% of cities were classified as the "intermediate coordination—advanced development" type, 26.95% as the "primary coordination—advanced development" type, and 14.54% as the "intermediate coordination—synchronous development" type. A total of 184 cities were identified as having advanced development,

accounting for more than 65%. It is evident that although the efficiency of GLUUL is consistently improving, HED remains relatively advanced, indicating that rapid development is underway in these cities and that their focus is still on the economy.

Fourthly, the influence degree and direction of Edl, Gtia, Isu, Fis, Ups, and Fs on CCD in different regions were significantly diverse. In 2020, Edl, Gtia, Isu, Ups, and Fs primarily contributed to the enhancement of CCD. Among them, Isu had the most significant promotion effect compared to other factors, and it had the strongest influence on the Yangtze economic corridor and the Beijing–Tianjin–Hebei region. The hindrance effect of Fis is currently impeding the CCD of the Northeast and western regions.

## 7. Discussion

Urban land as a natural resource with complex characteristics, and scholars have studied the relationship between it and economic development from the perspective of unilaterally influencing. Combined with related research, this article summarized and proposed the coupling coordination mechanism between GLUUL and HED, considering their richer connotations. In constructing the evaluation index system of urban land use efficiency, it notes that there are two main situations in previous studies. The first is that only economic and social outputs are considered in the desired outputs, neglecting the important component of ecological outputs [69]. The second is that the non-expected outputs only consider environmental pollution, such as industrial waste emissions, without taking into account carbon emissions during the urban land use process [70]. To comprehensively measure the land use efficiency of cities under environmentally friendly and low-carbon constraints, this study includes ecological outputs (carbon sinks in urban green spaces) and undesired outputs (carbon emissions from urban construction land) in the evaluation index system simultaneously. GLUUL efficiency values in South China and Northwest China were found to be higher, while those in Southwest China and Central China were lower, which is consistent with the results of Xie et al. and Yu et al. [50,69]. Additionally, the cities with high CCD are mainly distributed in the vicinity of the "Hu Huanyong line" and the southeast coastal area, which has a strong connection with the distribution characteristics of GLUUL and HED. It shows that when studying the CCD of two systems, often a better development of one system will lead to an increase in the overall coupling coordination value [56]. When exploring the CCD of GLUUL and HED, the development types of each city based on the relative degree of the two systems were investigated, and then the results of the coupling coordination subtype of GLUUL and HED were derived. This not only clarifies the CCD of each city but also identifies which sub-system is leading in this coupled system. Moreover, in exploring the influencing factors, it considered that each city has different characteristics in economic and social conditions, and different cities will be affected by the same factor in different ways. Therefore, the GTWR model was employed to study how different cities are affected by different factors at different time points. From the detailed regression results obtained, a deep understanding of the driving factors of CCD was gained.

Based on the results and discussions, this article proposed the following related policy implications.

Tailored Development Path for Different Types of Coordination.

Synchronous development cities. This type of city can reasonably allocate the input of land and economy in the process of development and achieve better output. In the future, based on coordinated development, they should further bolster the input of technology and talents whilst accelerating the adjustment of industrial structure and promoting transformation and upgrading in the mode of economic development. Meanwhile, it is also necessary to enhance the urban greening level and strengthen environmental pollution control measures to further improve the efficiency of GLUUL.

Advanced development cities. With the advantages of a high level of HED, this type of city should strengthen its investment in GLUUL in the future, make good plans for land use, actively redevelop inefficient utilization of land, implement new land use modes,

and encourage intensive, environmentally friendly, and low-carbon land use. Accelerating the promotion of GLUUL by upgrading the industrial structure and innovating in green technologies needs to be accompanied by a more pronounced emphasis on protecting urban land resources.

Backward development cities. Under the condition of the high efficiency of GLUUL, this type of city should strive to transform this advantage into an economic growth advantage in the future. Such cities generally have a better ecological environment. On the one side, they can attract projects and funds based on regional ecological advantages; on the other side, they actively accept the radiation of surrounding cities with advanced development, taking the sustainable development path of a green economy.

Orientated by Sustainable Development Exploring the Improvement Path of CCD.

Firstly, green and low-carbon industrial transformation should be accelerated. Optimizing the urban land use structure through upgrading the industrial structure is imperative. The indicators of urban land use ought to prioritize low energy consumption, low pollution, and high-benefit industries, including high-technology industries, environmental protection industries, and service industries. To raise the city's industrial level and speed up economic development, the industries are also implementing green, low-carbon production GHG reduction, so that economic development and environmental conservation go hand in hand.

Secondly, improve the financial support system. Enterprises will be encouraged to invest and innovate in green technologies through tax breaks, subsidies, and incentives. Implement specific policies to provide financial support for the development of green and low-carbon industries, ensuring sufficient financial resources are available to promote green economic development. Meanwhile, the government could improve the appeal of urban areas and encourage investment and employment by providing public service facilities and improving the urban environment. This would foster the coordinated development of GLUUL and HED.

Thirdly, promote the innovation and application of green technology. The innovation and application of various green technologies encompassing clean energy, clean production technology, and low-carbon emission technology can both diminish costs and advance competitiveness among enterprises, whilst simultaneously enhancing urban environmental standards and building an ecological civilization city. Meanwhile, promoting the application of new technologies, such as green intelligent transportation systems, could elevate the quality of urban services and strengthen the competitiveness of cities.

There are still some shortcomings in this study that need to be further explored. First, this study solely focuses on the theoretical analysis of the coupling and coordination mechanism between GLUUL and HED. Future research can delve deeper into the interaction mechanism between them through empirical evidence, such as utilizing the PVAR model or constructing a DEA evaluation model of the coordination relationship. Second, this study only analyzes how the coupling coordination of GLUUL and HED develops and evolves. In the future, further investigation of CCD between them can be conducted from the perspectives of optimization and prediction; for instance, constructing a spatial Markov chain model of CCD to predict its evolutionary trends. Third, due to data restrictions, our study was limited to covering only a portion of the cities in China, with a significant number of cities in western China excluded from consideration. A follow-up study can further expand the research scope and explore the changing trend of east–central–west in a more comprehensive way.

**Author Contributions:** Conceptualization, R.D. and J.L.; data curation, J.Z. and Y.H.; formal analysis, K.W.; visualization, W.X.; writing—original draft preparation, L.P.; writing—review and editing, L.P. and R.D. All authors have read and agreed to the published version of the manuscript.

**Funding:** This research was funded by the Guizhou University humanities and social science research project (No. 2023GZGXRW162), and the Guizhou Provincial Science and Technology Plan Project under Grant (No. QKHJC-ZK [2021] YB343).

**Institutional Review Board Statement:** Not applicable.

**Informed Consent Statement:** Not applicable.

**Data Availability Statement:** All data used in this paper are available in the city almanac, local statistical almanac, and EPS database. $CO_2$ emissions were calculated from the relevant conversion factors provided by the IPCC 2006 China Greenhouse Gas Inventory Guidelines, the baseline emission factors for each regional power grid, and the statistical data on centralized heat supply in each city over the years provided by the Statistical Yearbook of Urban Construction in China.

**Acknowledgments:** The authors are grateful to the statisticians for providing the data and to the editors and anonymous reviewers for their suggestions and comments.

**Conflicts of Interest:** The authors declare no conflicts of interest.

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
