# Peer review of "Exploring Spatial-Temporal Coupling and Its Driving Factors of Green and Low-Carbon Urban Land Use Efficiency and High-Quality Economic Development in China"

_sustainability, doi:10.3390/su16083455_

Round 1

Reviewer 1 Report

Comments and Suggestions for Authors

The article analyzes the spatiotemporal characteristics and coupling coordination relationship of green and low carbon use of urban land and high quality economic development in 282 cities in China, and studies the influencing factors of coupling coordination degree. The overall idea is clear, the methods are clear, and the research results are relatively reliable. However, there are issues with the production of graphics and the content of the article, which require some degree of modification. The specific suggestions are as follows:

1. The second and third paragraphs of the introduction both introduce research content related to land use efficiency, but GLUUL focuses on the urban land system, and the introduction does not focus on the research content of this article. It is recommended to modify it.

2. The fourth paragraph of the introduction is disconnected from the previous content. It is suggested to modify it to introduce the relationship between HED and land use, as well as the progress of related interdisciplinary research content, to support the research significance of this article.

3. Line 99: Only studying the coupling and coordination relationship between land resources and economic systems is not sufficient to promote the coupling and coordination of "resource-economy-environment". It is recommended to make modifications.

4. There are some shortcomings in the indicator system, such as Line 111: establishing an evaluation indicator system is not enough to be called indicator system innovation; Line 180: "It uniqueness lies in the comprehensive consideration of economic, social, and ecological output indicators, to quantify the maximization of social, economic, and ecological output and minimizing environmental loss in the urban land use course."; Lines 184-187: The input index includes labor factors, land factors, and capital factors, but each factor only has one data representative, and the indicator system is slightly weak. Suggest improving the relevant content.

5. Line 266: The "Coupling Coordination Degree" in Table 3 cannot explain the situation of 0.10<D<0.101 according to the grading standards in the table. It is recommended to modify it. In addition, it is recommended to unify the scale of coupling co scheduling in Table 3.

6. Line 286: "This paper finally identified 282 cities through China as the study object." It is suggested to provide specific explanations on the specific reasons for selecting the study area. Most of the eastern cities were selected as the study area, and what were the reasons for not being selected. In addition, the selected cities in the research area are mostly concentrated in the eastern part of China, which lacks comprehensiveness and cannot correspond to the title "Chinese cities". It is recommended to make modifications.

7. It is suggested to explain in the text what the grading criteria for Figures 4 and 5 are. Additionally, Figure 8 suggests explaining the special values section in the figure.

8. Line 418: "The cities at the bottom are located primarily in northern, central, and 418 southern China, including Yuncheng, Linfen, Hulunbuir, Shangqiu, Xiaogan, Shaoguan, and Hechi, etc." It is recommended to clarify the specific locations of these cities and keep them consistent with the context of the writing logic.

9. The description in section 4.2.3 is confusing, and it is unclear where the specific city being described is. It has no practical significance and should be coordinated with spatial location, such as its location throughout the country and direction. In addition, the previous studies only focused on four cross-sectional years, but 4.2.3 only studied 2020. It is recommended to supplement the reasons to facilitate the transition.

10. Lines 754-775: The article does not scientifically and quantitatively classify Chinese cities into different types. When comparing data in the previous text, it was based on seven geographical divisions. However, when making policy recommendations, it was divided into three types of urban development. The three types of urban development in the following text were not summarized based on the research results in the previous text, and the correspondence between the content in the previous text is not strong. It is recommended to reorganize and modify them.

11. Section 6.3: It is suggested to add more future development directions. Currently, in the discussion section of the article, the uniqueness of the research cannot be seen, and all are universal policies. It is suggested to make revisions.

12. There are some shortcomings in the production of graphics, such as the unclear reference between "border" and straight lines in the legend of Figure 2. It is recommended to make adjustments. Line 295: "Seven geographical administrative divisions were taken as the dividing standard..." It is recommended to explain in the text which of the seven geographical administrative regions are. In addition, although the article introduces seven geographical administrative divisions as the division criteria, Figure 2 does not reflect the seven regions, which poses a significant obstacle to readers' understanding. Please modify Figure 2 to address this issue. Why use two colors for the correlation of the same symbol in Figure 3? It is recommended to explain. It is recommended to use spatial distribution maps for Figures 6 and 7, as the existing drawings do not express it clearly. In the lower left corner of each sub graph of Figure 5, 8, 10-15 (ten segment enlarged view), the part containing the Chinese Mainland region should be colored, not black and white. It is recommended to modify it. For example, in the lower left corner of Figure 4 sub graph A, the Chinese Mainland region is colored. There is information obscured in Figure 6, please check and modify it. Figures 8-15 suggest adding labels to the key urban areas mentioned in the analysis.

13. Details:

(1) Suggest adding research periods in the abstract and Introduction .

(2) Suggest adding reference sources for data in Lines 41-42. Line 79: "The concept of HED originated in China,..." involves a description of the origin, and it is recommended to add corresponding references. In addition, Lines 298-316: This section contains many statements related to China's social development and other national conditions, and international readers outside of China are not clear about the socio-economic situation in various regions of China. Suggest adding references or supplementing descriptions of the social and economic backgrounds of each region to these related contents.

(3) Line 54: What kind of environmental losses does "environmental losses" specifically refer to, and whether it is appropriate to directly summarize it as environmental losses? Please provide detailed explanations.

(4) Lines 55-60: "In 2007, the 13th Conference of the Parties of the United Nations Framework Convention on Climate Change adopted the" Bali Action Plan ", which proposed the concepts of" low carbon economy "and" clean energy "and called on all countries to take measures to address global climate change." "Since then, more and more attention has been placed at how to achieve green and low carbon transformation of land utilization and raise the efficiency resource use" is recommended to be placed at the beginning of this chapter. In addition, Lines 528-532: "Divided into five levels using the natural break point method: strong, strong, medium, weak, and weak, which were displayed in different colors on the map. The meaning of the different colors is that: blue is weak, green is weak, yellow is medium, orange is strong, red is strong." It is recommended to include it in the research methods section.

(5) Line 94: "Few researchers integrate both" green "and" low carbon "into urban land use system," where "green" is a grammatical habit in Chinese, it is recommended to check English grammar.

(6) Line 108: "The influencing factors of CCD (GTWR)" in Figure 1 should not be considered as a part of the technical roadmap. It is recommended to align it with the context, such as "Analysis of influencing factors based on GTWR". In addition, there is no logical relationship between policy impact and research expectations, and it is recommended to remove the arrow connection.

(7) Line 169: "HED can positively promote GLUUL through scale, technology, and structural effects." What models, technologies, and structural effects can HED use to improve GLUUL? Please provide complete information. The same applies to GLUUL in the following sentence.

(8) There are some shortcomings in the format of the paper, such as Line 222:In Eq (2) And (3), inconsistent with the indentation of other explanatory formula paragraphs, it is recommended to unify. Line 247: The citation format of reference "(Tang, 2015)" is different from other references. It is recommended to modify and check for similar issues. Line 935: There are no references in this line, it is recommended to make revisions.

(9) Each level three subheading in section 4.2 is slightly longer, and it is recommended to modify it.

Comments on the Quality of English Language

Moderate editing of English language required

Author Response

Dear Reviewer,

    Thank you very much for your rigorous and detailed review to help us improve this paper. We value all your suggestions and appreciate the time and effort you have invested. We have revised the corresponding content according to your suggestions and make the following reply explanation, please see the attachment.

Reviewer 2 Report

Comments and Suggestions for Authors

This article presents potentially interesting information concerning spatiotemporal coupling and its driving factors of green and low-carbon land use efficiency and high-quality economic development in Chinese cities. This is a very interesting topic. The article is suitable for publication in a revised and improved form. My concerns are summarized below.

(1) The abstract section is incomplete because the beginning does not indicate why the research should be conducted, rather than just indicating what to do. Make clear what is new in theory and or methods used.

(2) The statement (Line 41-42) in the introduction section needs to indicate the data source.

(3) The article uses the abbreviation GLUUL multiple times, and it would be perfect if the full name could be listed in the introduction when the word is first used.

(4) Figure 3 seems to have missed the explanation for the division of China's four major regions. Please provide additional explanation.

(5) The discussion section needs to be further deepened, combined with result analysis and comparative analysis of existing research, to point out the contribution of this study.

(6) The conclusion (Line720-752) can be presented as a separate section.

Comments on the Quality of English Language

Minor editing of English language required.

Author Response

(The authors gave the same response as above.)

Reviewer 3 Report

Comments and Suggestions for Authors

L 38-39. Following papers are suggested to strengthen and enhance the statements. "Evaluating trends, benefits, and risks of global cities in recent urban expansion for advancing sustainable development" in Habitat International…"Mapping global urban land for the 21st century with data-driven simulations and Shared Socioeconomic Pathways" in Nat. Commun…

GLUUL pertains to the green and low-carbon utilization of urban land. To clarify, what constitutes urban land? Do you believe that rural and agricultural land within a city should also be categorized as urban land? In China, the term 'urban area' does not directly encompass the entire city; rather, it specifically refers to residential areas."

L335-338 could be revised as follows: 'Spatial-temporal analysis of GLUUL efficiency. Panels A, B, C, and D depict the changes in GLUUL efficiency in 2005, 2010, 2015, and 2020, respectively. The caption for similar figures could be altered like this.

For Fig. 4 and Fig. 5, how can we validate these results? For instance, providing data and the calculation process for example cities in another file would elucidate why certain cities obtained specific values. This would enable us to comprehend why certain cities, such as those in the northwest, achieved high GLUUL efficiency, while others like Chongqing attained lower values.

How can we validate or assess all the results presented in Fig. 5 and Fig. 8? Are there any references or additional data available?

Do you believe the GTWR regression model is suitable for analyzing the factors? Comparative methods are recommended for consideration.

Is it appropriate to include the conclusion in the discussion section?

Comments on the Quality of English Language

English very difficult to understand/incomprehensible

Author Response

Dear Reviewer,

    Thank you very much for your rigorous and detailed review to help us improve this paper. We value all your suggestions and appreciate the time and effort you have invested. We have revised the corresponding content according to your suggestions and make the following reply explanation.

Point to point response:

1. L 38-39. Following papers are suggested to strengthen and enhance the statements. "Evaluating trends, benefits, and risks of global cities in recent urban expansion for advancing sustainable development" in Habitat International…"Mapping global urban land for the 21st century with data-driven simulations and Shared Socioeconomic Pathways" in Nat. Commun…

Response: Thanks to your suggestion, we have read both documents carefully and strengthen the statements by quoting them. Please see the Line 49 in the revised version.

2. GLUUL pertains to the green and low-carbon utilization of urban land. To clarify, what constitutes urban land? Do you believe that rural and agricultural land within a city should also be categorized as urban land? In China, the term 'urban area' does not directly encompass the entire city; rather, it specifically refers to residential areas."

Response: Thank you for your question, which made us realize that explaining the composition of urban land in the text is very necessary. According to the Urban Land Use Classification and Planning and Construction Land Use Standards (GB50137-2011) of China, urban construction land includes residential land, land for public administration and public service facilities, land for commercial service facilities, industrial land, land for logistics and warehousing, land for roads and transportation facilities, land for public utilities, and land for green spaces and squares within the city, which constitute the urban land in this article, while the rural and agricultural land within cities should not be categorized as urban land. The construction of our indicators also corresponds to this. The “Urban built-up area” was selected as the input element, which is the area of actual constructed land. We have explained the composition of urban land in 3.1 Index system construction. Please see the Lines 241-246 in the revised version.

3. L335-338 could be revised as follows: 'Spatial-temporal analysis of GLUUL efficiency. Panels A, B, C, and D depict the changes in GLUUL efficiency in 2005, 2010, 2015, and 2020, respectively. The caption for similar figures could be altered like this.

Response: Thank you very much for your suggestion, which will make the figure captions in this paper more concise and clearer. We have revised the figure captions for all the spatial-temporal distribution maps according to your advice.

4. For Fig. 4 and Fig. 5, how can we validate these results? For instance, providing data and the calculation process for example cities in another file would elucidate why certain cities obtained specific values. This would enable us to comprehend why certain cities, such as those in the northwest, achieved high GLUUL efficiency, while others like Chongqing attained lower values.

Response: Thanks for your advice. For the case in Figure 4 where GLUUL efficiency is high in the Northwest China and low in Chongqing, we have provided basic data in 2020 of these cities in Sheet 1 of Attachment to help you validate the results. The case in Figure 5 that cities attained high HED levels located in eastern region while cities had a low HED levels lied in western regions, we have also provided relevant data in Sheet 2 of Attachment. The calculation process is consistent with that presented in Methods sections 3.2.1 and 3.2.2.

5. How can we validate or assess all the results presented in Fig. 5 and Fig. 8? Are there any references or additional data available?

Response: Thank you for your input. The results presented in Figure 5 we have provided data for your validation in Sheet 2 of Attachment. The results presented in Figure 8 were calculated based on the measured values of GLUUL efficiency and HED levels, and the calculation process is consistent with that introduced in Methods section 3.2.3. For your reference, we have provided part of the data in Sheet 3 of Attachment.

6. Do you believe the GTWR regression model is suitable for analyzing the factors? Comparative methods are recommended for consideration.

Response: Thanks for your question, we think it is suitable for analyzing the factors by the GTWR model. First, it can provide data test results (such as Table 8) to help us identify the goodness of fit of the model. On this basis, it also has the following two advantages compared with the econometric model and the Obstacle model: (1) It can provide detailed regression results from the dual dynamic perspectives of time and space [Huang, B.; Wu, B.; Barry, M. Geographically and temporally weighted regression for modeling spatio-temporal variation in house prices. International journal of geographical information science 2010, 24(3), 383-401.]. (2) Due to the differences in economic development and social conditions in different regions, the regression results based on the GTWR model are closer to the actual situation of each region [Zhao, Y.; Wang, Y.; Wang, Y. Comprehensive evaluation and influencing factors of urban agglomeration water resources carrying capacity. Journal of Cleaner Production 2021, 288, 125097]. This will assist local governments in formulating development strategies that are best suited to their unique circumstances based on the varying impacts of different factors.

7. Is it appropriate to include the conclusion in the discussion section?

Response: Thank you for making us rethink this, we realized that it is not appropriate to include the conclusions in the discussion section. We have presented them as a separate section. Please refer to the Lines 901-935 in the revised version.

Round 2

Reviewer 1 Report

Comments and Suggestions for Authors

1.The "Coupling Coordination Degree" in Table 3 cannot explain the situation of 0.40<D<0.401 based on the grading standards in the table.

2. The selected cities for the research area are mainly concentrated in the eastern part of China, which lacks comprehensiveness and cannot correspond to the title "Chinese cities". It is recommended to make modifications. The research area (as shown in the figure below) is mainly concentrated in the east, which cannot meet the requirements of comprehensiveness.

3.The classification criteria explained by Lines 429-437 are not comprehensive, and the levels of data such as 0.1982 and 0.2943 are missing.

4. Line 66: "Environmental losses" specifically refers to what kind of environmental losses, and can be directly summarized as whether environmental losses are appropriate? Please provide detailed explanations. The explanation states that it is not suitable to directly summarize it as environmental loss, but the text has not been deleted, and in the new manuscript, this section is on line 66. However, the explanation mentions lines 241-246, so please take it more seriously.

Comments on the Quality of English Language

Minor editing of English language required

Author Response

Dear Reviewer,

Thank you for the time and effort you have invested! We have revised the corresponding content according to your suggestions and make the following responses, please see the attachment.

Reviewer 3 Report

Comments and Suggestions for Authors

The paper is well revised

Author Response

Dear reviewer,

     Thank you so much for recognizing our paper and for all your hard work during the review process.